# Minimax Optimal Regret Bound for Reinforcement Learning with Trajectory Feedback

**Zihan Zhang** [1]   **Yuxin Chen** [2]   **Jason D. Lee** [3]   **Simon S. Du** [1]   **Ruosong Wang** [4]

## Abstract

In this work, we study reinforcement learning (RL) with trajectory feedback. Compared to the standard RL setting, in RL with trajectory feedback, the agent only observes the accumulative reward along the trajectory, and therefore, this model is particularly suitable for scenarios where querying the reward in each single step incurs prohibitive cost. For a finite-horizon Markov Decision Process (MDP) with $S$ states, $A$ actions and a horizon length of $H$, we develop an algorithm that enjoys an asymptotically nearly optimal regret of $\tilde{O}\left(\sqrt{SAH^3K}\right)$ in $K$ episodes. To achieve this result, our new technical ingredients include (i) constructing a tighter confidence region for the reward function by incorporating the RL with trajectory feedback setting with techniques in linear bandits and (ii) constructing a reference transition model to better guide the exploration process.

## 1. Introduction

In the standard reinforcement learning (RL) framework, it is assumed that the agent acts in an unknown environment, and in each step, the agent receives feedback in the form of a state-action dependent reward signal, and then transits to the next state. Although such an interaction model might be reasonable when a simulator is available, for real-life applications, such feedback model could be hard to realize. For practical scenarios, querying the reward function could be costly, or even impossible in certain circumstances.

As a motivating example, in healthcare, a doctor repeatedly interacts with a patient for the purpose of treatment. In each step, the doctor decides an action (e.g., taking certain medicines) and observes the new state (e.g., information like body temperature or blood pressure). On the other hand, the state-action dependent reward signal could be costly to observe, since the extent to which the disease has been cured might be expensive to measure as it requires comprehensive medical tests. In this case, in order to apply the RL framework, it is more reasonable to assume that the agent observes only the current state in each step, and the cumulative reward value is revealed only after a whole trajectory is finished.

As another example (which was also discussed in prior work on RL with trajectory feedback (Efroni et al., 2021)), in autonomous car driving, defining a state-action dependent reward function could be a challenging task, as it requires associating all possible state-action pairs with a real number. A possible workaround is to have human experts involved to produce the reward signals. However, defining reward signals could be a highly subjective matter, and waiting for reward values from human experts could take unacceptable amount of time for RL algorithms.

To circumvent issues mentioned above, practitioners often rely on heuristics (e.g., reward shaping (Ng et al., 1999)). RL with trajectory feedback has been recently proposed by Efroni et al. (2021) as a more principled framework. In this framework, the agent no longer has access to a per state-action reward function. Instead, it receives the cumulative reward on the trajectory as well as all the visited state-action pairs. Clearly, this new feedback model is harder than the standard RL setting and could be more applicable for real-life scenarios like healthcare or autonomous driving as mentioned in previous paragraphs. In the work by Efroni et al. (2021), algorithms based on the principle of optimism and Thompson sampling were proposed for RL with trajectory feedback. Although the algorithms in (Efroni et al., 2021) achieve $\sqrt{K}$-type regret bounds, the dependence on the number of states is far from being asymptotically optimal, [1] and obtaining asymptotically nearly optimal regret bounds

[1]Paul G. Allen School of CSE, University of Washington [2]Wharton Statistics and Data Science, University of Pennsylvania [3]Department of ECE, Princeton University [4]CFCS and School of Computer Science, Peking University. Correspondence to: Ruosong Wang <ruosongwang@pku.edu.cn>.

*Proceedings of the 42nd International Conference on Machine Learning*, Vancouver, Canada. PMLR 267, 2025. Copyright 2025 by the author(s).

---

[1]In this paper, by asymptotically nearly optimal, we mean that the regret bound is optimal up to logarithm factors when the number of episodes $K$ approaches infinity, which is standard in the RL theory literature (cf. (Azar et al., 2017; Agarwal et al., 2020)).

in this setting is the main focus of the present paper.

**Our contribution.** In this paper, we prove the first asymptotically nearly optimal regret for RL with trajectory feedback. Our main result is summarized as follows.

**Theorem 1.1** (Informal version of Theorem 5.6). *Fix $\delta > 0$. There exists an algorithm (Algorithm 1) such that for any episodic MDP with trajectory feedback, with probability $1 - \delta$, the regret in $K$ episodes is upper bounded by $\tilde{O}\left(\sqrt{SAH^3K}\right)^2$ for sufficiently large $K$.[3] Here $S$ is the number of states, $A$ is the number of actions, $H$ is the horizon length, and $K$ is the total number of episodes.*

It is known that for episodic MDPs, even if the agent has access to the per state-action reward function, the regret bound of any RL algorithm is lower bounded by $\Omega(\sqrt{SAH^3K})$ (Jin et al., 2018; Domingues et al., 2021). [4] Thus, the regret bound in Theorem 1.1 has nearly optimal dependence on $S$, $A$ and $H$ as $K$ approaches infinity, and therefore, our regret bound is asymptotically nearly optimal.

Conceptually, Theorem 1.1 shows that RL with trajectory feedback, a seemingly harder setting, has the same asymptotically optimal regret bound as the standard RL setting. Therefore, at least statistically, RL with trajectory feedback is no harder than the standard setting.

**Why trajectory-feedback is no harder.** It is well-known that in tabular MDPs, learning the transition kernel is harder than learning the reward, and RL algorithms usually pay more for learning the transition kernel. As a result, if the price of learning the rewards in RL with trajectory feedback is upper bounded by that of learning the transition kernel, RL with trajectory feedback would have the same asymptotically optimal regret bound as the standard RL setting. As will be shown later in Section 4, the regret incurred by learning the reward is indeed upper bounded by that of learning the transition kernel, and therefore, RL with trajectory feedback is no harder than the standard setting statistically.

**Computational efficiency.** Despite achieving an asymptotically nearly optimal regret bound, the algorithm for achieving Theorem 1.1 is not computationally efficient as it requires maintaining a set of deterministic policies whose cardinality could be exponential in $S$ and $H$, and an intriguing open problem is to design computationally-efficient algorithms for RL with trajectory feedback with asymptot-

---

[2] Throughout, we use $\tilde{O}(\cdot)$ to suppress logarithmic factors.

[3] As our main focus is to obtain asymptotically nearly optimal regret bounds, here we consider the case that $K$ approaches infinity while $S$, $A$ and $H$ are all fixed.

[4] In fact, the regret lower bound proved by Jin et al. (2018) is $\Omega(\sqrt{SAH^2T})$ with $T = KH$, which would be translated to $\Omega(\sqrt{SAH^3K})$ using our notations.

ically nearly optimal regret bounds, or show that such an algorithm does not exist.

## 2. Related Work

**RL with limited feedback.** As mentioned in the introduction, RL with trajectory feedback was first introduced by Efroni et al. (2021). Compared to the results in Efroni et al. (2021), our regret bound is asymptotically nearly optimal, while the results in Efroni et al. (2021) are not. In Section 4, a detailed comparison with the results in Efroni et al. (2021) from a technical perspective will be provided.

Cohen et al. (2021) designed an algorithm with $\sqrt{K}$-type regret bound that works for RL with trajectory feedback even when the noise is adversarially chosen. However, the regret bound by Cohen et al. (2021) is not asymptotically nearly optimal. Chatterji et al. (2021) considered a more general setting, where the reward revealed to the learner is no longer the cumulative reward on the sampled trajectory, but instead drawn from a logistic model. It is an interesting future direction to generalize our techniques to their setting and obtain asymptotically nearly optimal regret bounds.

Very recently, Cassel et al. (2024) considered RL with trajectory feedback in linear MDPs (Yang & Wang, 2019; Jin et al., 2020) and achieved a regret bound of $\tilde{O}(\sqrt{d^5H^7K})$. Translating their regret bound to the tabular setting considered in the present paper, the regret bound would be $\tilde{O}(\sqrt{S^5A^5H^7K})$ which is far from being asymptotically optimal. It would be interesting to generalize our techniques to RL with trajectory feedback when function approximation schemes are used and obtain improved regret bounds.

Preference-based RL (PbRL) (Wirth et al., 2017) is another RL paradigm to deal with the lack of reward signals. In PbRL, the agent receives feedback in terms of preferences over a trajectory pair instead of numerical rewards values. Compared to RL with trajectory feedback, PbRL is conceptually harder due to the lack of numerical feedback, and indeed, existing regret bounds for PbRL in the tabular setting (Novoseller et al., 2020; Xu et al., 2020b; Saha et al., 2023) is much worse than the nearly optimal regret bound obtained in this paper. PbRL was also studied in function approximation settings (Chen et al., 2022; Wu & Sun, 2023; Wang et al., 2023) and the bandit setting (Yue et al., 2012; Falahatgar et al., 2017; Bengs et al., 2021; Xu et al., 2020a).

**Linear bandits.** Linear bandit is a classical setting for modeling sequential decision-making problems, and various sample complexity bounds and regret bounds have been obtained in this setting and its generalizations (Dani et al., 2008; Abbasi-Yadkori et al., 2011; Li et al., 2019; Filippi et al., 2010; Li et al., 2019). We refer readers to Lattimore & Szepesvári (2020) for a comprehensive survey on this

topic. As observed in Efroni et al. (2021), there is a deep connection between RL with trajectory feedback and linear bandits. More specifically, RL with trajectory feedback can be understood as an instance of linear bandits over a convex set. Such a connection is also exploited in the present paper which will be discussed in more details in Section 4.

**Regret bounds in the standard RL setting.** There is a long line of work studying regret minimization in RL (Kakade, 2003; Jaksch et al., 2010; Azar et al., 2017; Jin et al., 2018; Zanette & Brunskill, 2019; Zhang & Ji, 2019; Zhang et al., 2020; Li et al., 2021; Zhang et al., 2022b; 2024). In particular, an asymptotically nearly optimal regret upper bound of $\tilde{O}\left(\sqrt{SAH^3K}\right)$ has been known in the literature (Azar et al., 2017), and more recent work typically focuses on the lower order terms, i.e., by considering the case where the total number of episodes $K$ is not that large compared to the number of states $S$, the number of actions $A$ and the horizon length $H$. In particular, the most recent work by Zhang et al. (2024) shows that an upper bound of $\tilde{O}\left(\sqrt{SAH^3K} + KH\right)$ can be achieved for any $K \geq 1$.

In this paper, to learn the transition kernel, we use a policy elimination framework similar to that in Zhang et al. (2022b). Compared to our algorithm, the algorithm in Zhang et al. (2022b) is designed for the standard RL setting and does not require the tighter confidence region construction for rewards. Such confidence region construction is the main technical contribution of the present paper.

## 3. Preliminaries

Throughout this paper, for an integer $N \geq 1$, we use $[N]$ to denote the set $\{1, 2, \ldots, N\}$.

**Episodic RL with trajectory feedback.** An MDP is defined as $M = \langle \mathcal{S}, \mathcal{A}, R, P, \mu \rangle$, where $\mathcal{S}$ is the state space, $\mathcal{A}$ is the action space, $R = \{R_h(s,a)\}_{(s,a) \in \mathcal{S} \times \mathcal{A}, h \in [H]}$ is the unknown reward distribution, $P = \{P_{s,a,h}\}_{(s,a) \in \mathcal{S} \times \mathcal{A}, h \in [H]}$ is the unknown transition kernel and $\mu$ is the initial distribution. For a state-action pair $(s,a) \in \mathcal{S} \times \mathcal{A}$, a level $h \in [H]$ and state $s' \in \mathcal{S}$, $P_{s,a,h,s'}$ is the probability of transiting from $s$ to $s'$ at level $h$ after taking action $a$. We assume that the reward distribution $R_h(s,a)$ is supported on $[0,1]$ with mean $R_h(s,a)$.

A (deterministic) policy $\pi$ can be viewed as a collection of mappings $\{\pi_h\}_{h=1}^H$ where each $\pi_h : \mathcal{S} \rightarrow \mathcal{A}$ is a mapping from the state space to the action space. We use $\Pi_{\text{det}}$ to denote the set of all deterministic policies. We also consider mixtures of deterministic policies which can be seen as a distributions over $\Pi_{\text{det}}$. A policy $\pi$ induced a random trajectory $(s_1, a_1, s_2, a_2, \ldots, s_H, a_H)$, where $s_1 \sim \mu$, $a_h = \pi_h(s_h)$ and $s_{h+1} \sim P_{s_h, a_h, h}$ for all $h \in [H]$.

In each episode, the agent decides a policy and observes the induced trajectory $(s_1, a_1, s_2, a_2, \ldots, s_H, a_H)$. At the end of each episode, the agent also receives a trajectory reward feedback $Y = \sum_{h=1}^H r_h$, where each $r_h$ is independently drawn from $\mathcal{R}_h(s_h, a_h)$. We use $\mathcal{T}$ denote the set of all possible trajectories. Note that $|\mathcal{T}| = (SA)^H$.

For a policy $\pi$, the $Q$-function and $V$-function are given by

$$Q_h^\pi(s,a) = \mathbb{E}_\pi \left[ \sum_{h'=h}^H r_{h'}(s_{h'}, a_{h'}) \Big| (s_h, a_h) = (s,a) \right];$$

$$V_h^\pi(s) = \mathbb{E}_\pi \left[ \sum_{h'=h}^H r_{h'}(s_{h'}, a_{h'}) \Big| s_h = s \right].$$

The optimal $Q$-function and $V$-function are given by

$$Q_h^*(s,a) = \sup_{\pi \in \Pi_{\text{det}}} Q_h^\pi(s,a);$$

$$V_h^*(s) = \max_{\pi \in \Pi_{\text{det}}} V_h^\pi(s) = \max_a Q_h^*(s,a).$$

We use $\pi^*$ to denote an optimal deterministic policy[5] such that $Q_h^*(s,a) = Q_h^{\pi^*}(s,a)$ for all $(s,a,h)$. Moreover, for a given transition kernel $p$ and reward distribution $r$, define

$$W^\pi(r,p) = \mathbb{E}_{\pi,p} \left[ \sum_{h=1}^H r_h(s_h, a_h) \right];$$

$$W^*(r,p) = \max_{\pi \in \Pi_{\text{det}}} W^\pi(r,p).$$

Let $\pi^k$ be the policy executed by the agent in the $k$-th episode, the regret is defined by

$$\text{Regret}(K) := \sum_{k=1}^K \left( W^*(R,P) - W^{\pi^k}(R,P) \right).$$

**Other Notations.** Given a policy $\pi$ and a transition kernel $p$, we use $\mathbb{E}_{\pi,p}[\cdot]$ ($\text{Pr}_{\pi,p}[\cdot]$) to denote the expectation (probability) under the policy $\pi$ and transition kernel $p$. In particular, for a trajectory $\tau = \{(s_h, a_h)\}_{h=1}^H$, $\text{Pr}_{\pi,p}[\tau]$ is the probability of observing $\tau$ under $\pi$ and $p$. We also define the occupancy function $d_p^\pi(s,a,h) = \mathbb{E}_{\pi,p}[\mathbb{I}[(s_h, a_h) = (s,a)]]$. We use $d_p^\pi$ to denote the $SAH$-dimensional vector $\{d_p^\pi(s,a,h)\}_{(s,a,h) \in \mathcal{S} \times \mathcal{A} \times [H]}$. We may also regard $R$ as a $SAH$-dimensional vector $\{R_h(s,a)\}_{(s,a,h) \in \mathcal{S} \times \mathcal{A} \times [H]}$. Given a trajectory $\tau = \{(s_h, a_h)\}_{h=1}^H$, we let $\phi_\tau \in \mathbb{R}^{SAH}$ to be the vector such that $\phi_\tau(s', a', h) = \mathbb{I}[(s', a') = (s_h, a_h)]$. We use $\mathbf{I}$ to denote the $SAH$-dimensional identity matrix. We use $\mathcal{E}^C$ to denote the complement of the set $\mathcal{E}$.

## 4. Technical Overview

In this section, we give an overview of the technical challenges for obtaining the asymptotically nearly optimal regret

---

[5]It is well known that optimal $Q$-function and $V$-function can be achived by a deterministic policy.

bound for RL with trajectory feedback, together with our approaches to tackle these challenges. To explain the high-level ideas, we first consider the simpler setting that the transition kernel $P$ is known, and then switch to the general setting in which case the transition kernel is unknown.

**Connection with linear bandits.** As observed in prior work (Efroni et al., 2021), when the transition kernel $P$ is known, RL with trajectory feedback can be seen as an instance of linear bandits. More specifically, in each episode, suppose the trajectory observed by the agent is $\tau$, the expected trajectory reward feedback would be $\phi_\tau^\top R$, i.e., a linear function with respect to $\phi_\tau$. Based on this observation, Efroni et al. (2021) showed how to build appropriate confidence regions for RL with trajectory feedback by adapting analysis for linear bandits algorithm and obtained a regret bound of $\tilde{O}\left(\sqrt{S^2 A^2 H^4 K}\right)$. Although it is plausible to improve their regret bound to $\tilde{O}\left(\sqrt{S^2 A H^3 K}\right)$ by a more refined analysis, it is unclear how to improve the dependence on $S$ in their regret bound. Indeed, in the work by Efroni et al. (2021), RL with trajectory feedback is naïvely treated as an instance of linear bandits with feature dimension $d = SAH$, and the best known regret bound for any linear bandits algorithm is $\tilde{O}(d\sqrt{T})$ (Dani et al., 2008), or $O(\sqrt{dT \log K})$ for linear bandits with $K$ arms (Bubeck et al., 2012). Since there are $A^{SH}$ policies for an MDP, and each of them can be seen as an arm in the linear bandits problem instance, improving the order of $S$ in the regret bound of prior work requires fundamentally new ideas.

**Tighter confidence region based on trajectories.** In order to achieve a minimax optimal regret bound, our first new idea is to build a tighter confidence region by exploiting structures of the linear bandits instance associated with RL with trajectory feedback. Before getting into more details, we first review least squares regression (LSR), an estimator commonly used in linear bandits algorithms (also in prior work on RL with trajectory feedback (Efroni et al., 2021)).

Given a set of data points $\{\pi^t, \tau^t, Y^t\}_{t=1}^T$, where for each $1 \leq t \leq T$, $\pi^t$ is the policy executed in the $t$-th episode, $\tau^t$ is the trajectory sampled by executing $\pi^t$ and the $Y^t$ is the trajectory reward feedback. Clearly, $\mathbb{E}[Y^t] = \phi_{\tau^t}^\top R$, which motivates the design of the LSR estimator

$$\hat{R} = \arg\min_r \sum_{t=1}^T \left(Y^t - \phi_{\tau^t}^\top r\right)^2 + \lambda\|r\|_2^2 = \Lambda^{-1}\sum_{t=1}^T \phi_{\tau^t} Y^t,$$
$$(1)$$

where $\Lambda = \lambda \mathbf{I} + \sum_{t=1}^T \phi_{\tau^t}\phi_{\tau^t}^\top$ is the information matrix. Optimism-based linear bandits algorithms typically maintain a set of arms, and eliminate arms outside the confidence region during the execution of the algorithm. For RL with

trajectory feedback, each arm in the linear bandits instance corresponds to a deterministic policy in the original MDP.

Our construction of the tighter confidence region is based on the following two observations:

- Although the total number of deterministic policies could be as large as $A^{SH}$, the number of trajectories is $|\mathcal{T}| = (SA)^H$ which is much smaller than the number of deterministic policies;

- For any deterministic policy $\pi$, $d_P^\pi = \sum_{\tau \in \mathcal{T}} \Pr_{\pi,P}[\tau] \cdot \phi_\tau$ which is a convex combination of $\{\phi_\tau\}_{\tau \in \mathcal{T}}$.

Based on these observations, instead of building confidence region for $|(d_P^\pi)^\top(\hat{R} - R)|$ for each deterministic policy $\pi$ and applying a union bound over all policies which result in suboptimal regret bounds, we consider the following event

$$\mathcal{E} := \left\{\forall \tau \in \mathcal{T}, \left|\phi_\tau^\top(\hat{R} - R)\right| \right.$$
$$\left. \leq c\left(\min\left\{\sqrt{\phi_\tau^\top \Lambda^{-1}\phi_\tau \sigma^2 \log\left(\frac{|\mathcal{T}|}{\delta}\right)}, H\right\}\right)\right\}, \quad (2)$$

where $c$ is an absolute constant and $\sigma^2 \leq H$ is a constant such that $\{Y^t - \phi_{\tau^t}^\top R\}_{t=1}^T$ is a group of independent zero-mean $\sigma^2$-subgaussian random variables. By standard concentration arguments, $\mathcal{E}$ holds with probability at least $1 - \delta$. We assume $\mathcal{E}$ holds in the remaining part of the discussion.

Note that second observation mentioned above implies that for any policy $\pi$,

$$\left|(d_P^\pi)^\top(\hat{R} - R)\right|$$
$$\leq \sum_{\tau \in \mathcal{T}} \Pr_{\pi,P}[\tau]\left|\phi_\tau^\top(\hat{R} - R)\right|$$
$$\leq O\left(\sum_{\tau \in \mathcal{T}} \Pr_{\pi,P}[\tau] \min\left\{\sqrt{\phi_\tau^\top \Lambda^{-1}\phi_\tau H \log(2|\mathcal{T}|/\delta)}, H\right\}\right)$$
$$\leq \tilde{O}\left(H\sqrt{\sum_{\tau \in \mathcal{T}} \Pr_{\pi,P}[\tau] \min\{\phi_\tau^\top \Lambda^{-1}\phi_\tau, 1\}}\right), \quad (3)$$

where the last step holds by Cauchy-Schwarz inequality, the fact that $|\mathcal{T}| = (SA)^H$, and suppressing $\log(SA)$ factors into the $\tilde{O}(\cdot)$ notation.

**Exploration by optimal design.** During the execution of the algorithm, we maintain a set of deterministic policies $\Pi$ that have not been eliminated. According to (3), in order to prove a uniform upper bound for $\left|(d_P^\pi)^\top(\hat{R} - R)\right|$ for all deterministic policies $\pi \in \Pi$, it suffices to bound

$$\max_{\pi \in \Pi} \sum_{\tau \in \mathcal{T}} \Pr_{\pi,P}[\tau] \min\left\{\phi_\tau^\top \Lambda^{-1}\phi_\tau, 1\right\}. \quad (4)$$

For this purpose, we need to carefully choose a set of policies $\{\pi^t\}_{t=1}^T$, so that the quantity in (4) is upper bounded. As another new technical ingredient, we show how to generalize the classical Kiefer–Wolfowitz Theorem (see Lemma B.1) to our setting. In particular, in Lemma B.2 in the supplementary material, we show that there exists $\bar{\pi}$ which is a mixture of deterministic policies, such that

$$\max_{\pi \in \Pi} \sum_{\tau \in \mathcal{T}} \Pr_{\pi, P}[\tau] \phi_\tau^\top \Lambda_{\bar{\pi}}^{-1} \phi_\tau = SAH, \qquad (5)$$

where $\Lambda_{\bar{\pi}} := \sum_{\tau \in \mathcal{T}} \Pr_{\bar{\pi}, P}[\tau] \phi_\tau \phi_\tau^\top$. Therefore, by running $\bar{\pi}$ for $T$ steps, we could collect an information matrix $\Lambda \succcurlyeq cT\Lambda_{\bar{\pi}}$ with high probability for some absolute constant $c > 0$. Combining (3) and (5), we obtain that

$$\max_{\pi \in \Pi} \left| (d_P^\pi)^\top (\hat{R} - R) \right| \leq \tilde{O}\left( H\sqrt{SAH/T} \right). \qquad (6)$$

In summary, with the arguments above, for any policy set $\Pi$, we are able to collect a dataset $\{\pi^t, \tau^t, Y^t\}_{t=1}^T$ in $T$ episodes to obtain $\hat{R}$, such that

$$\max_{\pi \in \Pi} \left| W^\pi(\hat{R}, P) - W^\pi(R, P) \right| = \max_{\pi \in \Pi} \left| (d_P^\pi)^\top (\hat{R} - R) \right|$$
$$\leq \tilde{O}\left( \sqrt{SAH^3/T} \right). \qquad (7)$$

**Online batch learning by policy elimination.** Finally, we show how to combine the two technical ingredients mentioned above into the framework of online policy elimination. In this framework, the learning process is divided into consecutive batches. The algorithm maintains a policy set during its execution. Suppose the policy set maintained is $\Pi_\ell$ at the beginning of the $\ell$-th batch. The algorithm will eliminate a subset of policies from $\Pi_\ell$ to form $\Pi_{\ell+1}$ in the $\ell$-th batch. Initially, we set $\Pi_1$ to be the set of all deterministic policies. Moreover, there are $O(\log \log K)$ batches for the whole algorithm, and there are $K_\ell = 2K^{1 - \frac{1}{2^\ell}}$ episodes in the $\ell$-th batch.

As an invariant, during the execution of the algorithm, we always have that the optimal policy $\pi^* \in \Pi_\ell$ for all $\ell$. By (7), for each $\ell$, we obtain a set of estimated reward values $\hat{R}^\ell$ such that

$$\max_{\pi \in \Pi_\ell} \left| W^\pi(\hat{R}, P) - W^\pi(R, P) \right| \leq \tilde{O}\left( \sqrt{SAH^3/K_\ell} \right).$$

By setting

$$\Pi_{\ell+1} = \left\{ \pi \in \Pi_\ell : \max_{\pi' \in \Pi_\ell} W^{\pi'}(\hat{R}, P) - W^\pi(\hat{R}, P) \leq \epsilon_\ell \right\} \qquad (8)$$

where $\epsilon_\ell = \tilde{O}\left( \sqrt{SAH^3/K_\ell} \right)$, it holds that $\pi^* \in \Pi_{\ell+1}$ and

$$W^*(R, P) - W^\pi(R, P) \leq \tilde{O}\left( \sqrt{SAH^3/K_\ell} \right)$$

for any $\pi \in \Pi_{\ell+1}$. Therefore, the regret in the $(\ell+1)$-th batch is bounded by

$$\tilde{O}(K_{\ell+1}\sqrt{SAH^3/K_\ell}) = \tilde{O}(\sqrt{SAH^3 K}),$$

which means that the total regret is at most $\tilde{O}(\sqrt{SAH^3 K})$.

**Dealing with unknown transition kernels.** In the discussion above, we assume that the transition kernel $P$ is known. Now we discuss how to remove such an assumption by learning the transition kernel in an online fashion. In order to implement the elimination-based online batch learning process mentioned above, we only need the transition kernel (i) to design the exploration policy so that (5) is ensured and (ii) to ensure the policy elimination step in (8) can be accurately implemented.

To achieve (i) and (ii), we first obtain a reference transition kernel $\tilde{P}$, *which serves as an efficient tool to help design the exploration policy*. Following the regret analysis for online batch learning in (Zhang et al., 2022b), the regret stemming from learning $\tilde{P}$ can be bounded by $\tilde{O}(\sqrt{SAH^3 K})$ (with lower order terms ignored). Moreover, for (i), an exact solution for (5) is not necessary. Instead, an approximate solution with a constant competitive ratio is sufficient to guide the exploration process, which could be found with the assistance of a reference model. For (ii), by using samples obtained by executing the exploration policy found in (i), we could build an empirical transition kernel which would be sufficient for implementing the policy elimination step in (8).

**Computational efficiency.** Given an approximate transition kernel, the algorithm by Zhang et al. (2022b) achieves computationally efficient batch learning in the standard RL setting. In contrast, in the RL with trajectory feedback setting studied in this work, even with full knowledge of the transition kernel, we suffer from computational inefficiency due to the lack of reward information. In our algorithm, we maintain a subset of deterministic policies and solve optimization problems over the remaining policies (cf. Line 4 in Algorithm 3). Recall that initially we include all deterministic policies, and therefore the number of remaining policies could be exponential in $S$ and $H$ during the execution of the algorithm. To our best of knowledge, no existing algorithm could solve such optimization problem efficiently even if the transition kernel is known and approximation is allowed, and therefore, our algorithm is also not computationally efficient. We leave obtaining computationally efficient algorithm as an interesting future direction.

# 5. Algorithm

In this section, we present our algorithm. The specific choice of the parameters could be found in Appendix A. The main algorithm (Algorithm 1) has two stages.

The first stage (Line 3 in Algorithm 1) serves to acquire a coarse approximation $p$ of the transition kernel $P$, guiding the design of exploration policy. Instead of approximating $P$ to the $L_1$-norm, it is required the trajectory distribution under $P$ could be covered by that under $p$ up to a constant ratio. Formally, we have the definition below to measure the similarity between two transition kernels.

**Definition 5.1.** For two transition kernels $p$ and $p'$, we say $p$ is an $(n, x)$-approximation of $p'$ with respect to a set of policies $\Pi$ iff $\mathcal{S} \times \mathcal{A} \times \mathcal{S} \times [H]$ could be divided into two sets $\mathcal{K}$ and $\mathcal{K}^{\mathrm{C}}$ such that

$$e^{-\frac{\log(n)}{H}} p'_{s,a,h,s'} \le p_{s,a,h,s'} \le e^{\frac{\log(n)}{H}} p'_{s,a,h,s'},$$
$$\forall (s, a, h, s') \in \mathcal{K}; \quad (9)$$

$$\Pr_{\pi,p}[\mathcal{K}^{\mathrm{C}}] = 0, \quad \forall \pi \in \Pi_{\mathrm{det}}; \quad (10)$$

$$\max_{\pi \in \Pi} \Pr_{\pi,p'}[\mathcal{K}^{\mathrm{C}}] \le x, \quad (11)$$

where $\Pr_{\pi,q}[\mathcal{K}^{\mathrm{C}}]$ is the probability of visiting $\mathcal{K}^{\mathrm{C}}$ under policy $\pi$ and transition kernel $q$.

The second stage consists of several consecutive batches. In each batch of the second stage (Line 5 in Algorithm 1), we search for an exploration policy $\bar{\pi}$ by using the coarse model $p$ obtained during the first stage. Subsequently, we execute $\bar{\pi}$ to collect the trajectory feedback, and construct reward confidence region $\mathcal{R}$ with least squares regression to conduct policy elimination.

---

**Algorithm 1**

1: **Input:** total number of episodes $K$.
2: **Initialization:** Set $K_0, L, \{K_\ell\}_{1 \le \ell \le L}, \epsilon_0, \sigma_0, \kappa$ according to Appendix A;
3: $\{\tilde{P}, \Pi_1\} \leftarrow$ Ref-Model$(K_0, K)$;
4: **for** $\ell = 1, 2, \ldots, L$ **do**
5: $\quad \Pi_{\ell+1} \leftarrow$ Traj-Learning$(\tilde{P}, K_\ell, \Pi_\ell)$;
6: **end for**

---

## 5.1. Learning the Reference Model (the First Stage)

We present the algorithm for learning the reference model in Algorithm 2. The algorithm consists of four stages. Initially, the goal is to acquire a coarse reference model. In the subsequent stage, the focus shifts to learning a coarse reward estimator. The third stage involves gathering samples to execute policy elimination, ensuring that the remaining policies are approximately $O(\epsilon_0)$-optimal. In the final stage, we invoke Raw-Exploration with a larger length to obtain a more refined reference model.

**Raw exploration.** In Algorithm 2, we invoke Raw-Exploration (see Algorithm 6 in Appendix C) to learn a proper reference model. This algorithm is based on Algorithm 2 (Zhang et al., 2022b), with slight modification so that it could be applied to general policy set $\Pi$.

---

**Algorithm 2** Ref-Model$(K_0, K)$

1: **Input:** length $K_0$, total length $K$;
2: $\bar{K}_1 = 1000\sqrt{SAHK}$, $\bar{K}_2 = K_0 - 3\bar{K}_1$
3: $\hat{P}_1 \leftarrow$ Raw-Exploration$(\Pi_{\mathrm{det}}, \bar{K}_1)$;
4: $\hat{R} \leftarrow$ Reward-Regression$(P_1, \Pi_{\mathrm{det}}, \bar{K}_1)$;
5: $\Pi_1 \leftarrow$ Plan$(\hat{R}, \hat{P}_1, \bar{K}_1, \Pi_{\mathrm{det}}, \epsilon_0)$;
6: $\hat{P}_2 \leftarrow$ Raw-Exploration$(\Pi_1, \bar{K}_2)$;
7: **return:** $\{\hat{P}_2, \Pi_1\}$.

---

Properties of the learned model are summarized as below.

**Lemma 5.2.** *By running* Ref-Model$(K_0, K)$, *with probability* $1 - \delta$, *it holds that*

- $\hat{P}_2$ *is a* $(3, \sigma_0)$-*approximation of* $P$ *w.r.t.* $\Pi_1$;

- $\pi^* \in \Pi_1$;

- $W^\pi(R, P) \ge W^*(R, P) - 2\epsilon_0$ *for any* $\pi \in \Pi_1$.

The proof of Lemma 5.2 is postponed to Appendix D.1

## 5.2. Policy Elimination with Reward Regression (the Second Stage)

---

**Algorithm 3** Traj-Learning$(p, \check{K}, \Pi)$

1: **Input:** reference model $p$, length $\check{K}$, policy set $\Pi$;
2: $\hat{R} \leftarrow$ Reward-Regression$(p, \check{K}, \Pi)$;
3: $\Pi_{\mathrm{next}} \leftarrow$ Plan$(\hat{R}, p, \check{K}, \Pi, \kappa)$;
4: **return:** $\Pi_{\mathrm{next}}$.

---

We present the algorithm for reward regression and policy elimination in Algorithm 3, which has two parts. In the first part, the goal is to learn the reward by least squares regression, while the goal of the second part is to eliminate policies in $\Pi$ based on the reward learned in the first part.

Throughout Algorithm 3, the agent only executes policies within $\Pi$, and therefore, the regret incurred by running Algorithm 3 is upper bounded by $\check{K} \cdot \max_{\pi \in \Pi}(W^*(R, P) - W^\pi(R, P))$ where $\check{K}$ is the total number of episodes.

**Reward regression.** We compute the optimal design policy according to the reference model $p$, and then collect trajectory feedback to learn the reward function. It is worth noting that the least squares regression estimator $\bar{R}$ (see Line 12 in Algorithm 4) might escape $[0, 1]^{SAH}$, and thus we construct a reward confidence region $\mathcal{R}$ (see Line 13

**Algorithm 4** `Reward-Regression`$(p, \check{K}, \Pi)$

1: **Input:** reference model $p$, length $\check{K}$, policy set $\Pi$;
2: $\lambda \leftarrow \frac{1}{SAH^2\check{K}}, \Lambda \leftarrow \lambda\mathbf{I}, \check{K}_1 \leftarrow \frac{\check{K}}{54\log(\frac{2d}{\delta})}$
3: **for** $t = 1, 2, \ldots, \check{K}_1$ **do**
4:    $\pi^t \leftarrow \arg\max_{\pi \in \Pi} \sum_{\tau \in \mathcal{T}} \Pr_{\pi,p}[\tau] \cdot \min\{\phi_\tau^\top \Lambda^{-1}\phi_\tau, 1\}$;
5:    $\Lambda \leftarrow \Lambda + \sum_{\tau \in \mathcal{T}} \Pr_{\pi^t,p}[\tau]\phi_\tau\phi_\tau^\top \cdot \frac{1}{\max\{\phi_\tau^\top \Lambda^{-1}\phi_\tau, 1\}}$;
6: **end for**
7: $\bar{\pi}$ be the mixed policy which plays $\pi^t$ with probability $1/\check{K}_1$ for each $1 \leq t \leq \check{K}_1$;
8: **for** $t = 1, 2, \ldots, \check{K}$ **do**
9:    Run $\bar{\pi}$ to get trajectory $\tau^t$ and trajectory reward feedback $Y^t$;
10: **end for**
11: $\hat{\Lambda} \leftarrow 18\lambda\log(2d/\delta)\mathbf{I} + \sum_{t=1}^{\check{K}} \phi_{\tau^t}\phi_{\tau^t}^\top$;
12: $\bar{R} \leftarrow \bar{\Lambda}^{-1}\sum_{t=1}^{\check{K}} Y^t\phi_{\tau^t}$
13: $\mathcal{R} \leftarrow \{\tilde{R} \in [0,1]^{SAH} : |\phi_\tau^\top(\tilde{R} - \bar{R})|$
       $\leq 8\sqrt{H^2\log^2(\frac{4SAH}{\delta})\phi_\tau^\top\hat{\Lambda}^{-1}\phi_\tau}, \forall\tau \in \mathcal{T}\}$;
14: **if** $\mathcal{R} \neq \emptyset$ **then**
15:    Choose $\hat{R} \in \mathcal{R}$ ;
16: **else**
17:    $\hat{R} \leftarrow \mathbf{0}$;
18: **end if**
19: **return:** $\hat{R}$

---

in Algorithm 4) instead. For Algorithm 4, we have the following lemma to bound the error of reward regression.

**Lemma 5.3.** *If $p$ is a $(3, x)$-approximation of $P$ with respect to $\Pi$ for some $x \geq 0$, with probability $1 - \delta$, it holds that*

$$\max_{\pi \in \Pi}\left|W^\pi(\hat{R}, P) - W^\pi(R, P)\right|$$
$$\leq H\sqrt{\log^2\left(\frac{4SAH}{\delta}\right)} \cdot \left(x + 325\sqrt{\frac{SAH\log^2(\frac{2SAH\check{K}}{\delta})}{\check{K}}}\right)$$

*where $\hat{R} = $ `Reward-Regression`$(p, \check{K}, \Pi)$.*

**Policy elimination.** With the reward estimator $\hat{R}$, we proceed to construct the confidence region to facilitate policy elimination. As described in Algorithm 5, for every batch, we use the reference model $p$ and policy design to find an exploration policy $\bar{\pi}$ with nearly optimal coverage. By executing $\bar{\pi}$, we obtain an empirical transition model which is then used to eliminate policies. Formally, we have the uniform convergence guarantee for Algorithm 5.

**Lemma 5.4.** *Fix $x, y, z, \epsilon \geq 0$. Assume that*

- *$\pi^* \in \Pi$;*

- *$p$ is a $(3, x)$-approximation of $P$ w.r.t. $\Pi$;*

- *$W^\pi(u, P) \geq W^*(u, P) - y$ for any $\pi \in \Pi$;*

- *$\max_{\pi \in \Pi}|W^\pi(u, P) - W^\pi(R, P)| \leq z$;*

- *$\epsilon \geq 2(b + z)$, where*

$$b = 30\sqrt{\frac{SAH^2(H + Sy)\log\left(\frac{8SAH}{\delta}\right)}{\check{K}}}$$
$$+ \frac{360S^2AH^3\log\left(\frac{8SAH}{\delta}\right)}{\check{K}} + 4SAH^2x.$$

*Let $\Pi_{\text{next}} = $ `Plan`$(u, p, \check{K}, \Pi, \epsilon)$. With probability $1 - \delta$, it holds that:*

- *the optimal policy $\pi^* \in \Pi_{\text{next}}$;*

- *$W^\pi(R, P) \geq W^*(R, P) - 2\epsilon$ for any $\pi \in \Pi_{\text{next}}$.*

Lemma 5.4 states that, given some proper reference transition kernel $p$ and a policy set $\Pi$ such that $\pi^* \in \Pi$, Algorithm 5 could return a policy set with better confidence bounds by policy elimination.

---

**Algorithm 5** `Plan`$(u, p, \check{K}, \Pi, \epsilon)$

1: **Input:** reward function $u$, reference model $p$, episode length $\check{K}$, policy set $\Pi$, threshold $\epsilon$
2: $\bar{\pi} \leftarrow$ `Design`$(\Pi, p)$;
3: Execute $\bar{\pi}$ in the next $\check{K}$ episodes, and collect samples as $\mathcal{D}$;
4: $N_h(s, a) \leftarrow$ the count of $(s, a, h)$ in $\mathcal{D}$;
5: **for** $(s, a, h) \in \mathcal{S} \times \mathcal{A} \times [H]$ **do**
6:    $\hat{p}_{s,a,h} \leftarrow$ the empirical transition probability of the samples of $(s, a, h)$ in $\mathcal{D}$;
7: **end for**
8: $\Pi_{\text{next}} \leftarrow \left\{\pi \in \Pi : W^\pi(u, \hat{p}) \geq \max_{\pi' \in \Pi} W^{\pi'}(u, \hat{p}) - \epsilon\right\}$
9: **return:** $\Pi_{\text{next}}$.
'10: **Function:** `Design`$(\Pi, p)$;
11: Let $\lambda = \{\lambda_\pi\}_{\pi \in \Pi} \in \Delta^\Pi$ be the distribution

$$\lambda \leftarrow \text{argmin}_{\lambda' = \{\lambda'_\pi\}_{\pi \in \Pi} \in \Delta^\Pi} \max_{\pi^* \in \Pi} \sum_{s,a,h} \frac{d_p^{\pi^*}(s, a, h)}{\sum_\pi \lambda'_\pi d_p^\pi(s, a, h)};$$

12: **return:** $\bar{\pi}$ be the mixed policy which plays $\pi \in \Pi$ with probability $\lambda_\pi$;

---

Based on Lemma 5.3 and Lemma 5.4, we summarize the guarantees of Algorithm 3 as below.

**Lemma 5.5.** *Let $\iota = \log^2\left(\frac{16SAH\check{K}}{\delta}\right)$. Let $\Pi_{\text{next}} = $ `Traj-Learning`$(p, \check{K}, \Pi)$. Fix $\tilde{x}, \tilde{y}, \kappa \geq 0$. Assume that*

- *$\pi^* \in \Pi$;*

- *$p$ is a $(3, \tilde{x})$-approximation of $P$ with respect to $\Pi$;*

- $W^\pi(R, P) \geq W^*(R, P) - \tilde{y}$ for any $\pi \in \Pi$;

- $\kappa \geq 20\big(72\sqrt{\frac{SAH^3\iota}{\tilde{K}}} + 6\sqrt{\frac{S^2AH^2\tilde{y}\iota}{\tilde{K}}} + \frac{100S^2AH^3\iota}{\tilde{K}} + SAH^2\tilde{x}\iota\big)$.

With probability $1 - \delta$, it holds that $\pi^* \in \Pi$ and

$$W^\pi(R, P) \geq W^*(R, P) - 2\kappa$$

for any $\pi \in \Pi_{\text{next}}$.

The full proofs of Lemma 5.3, Lemma 5.4 and Lemma 5.5 are presented in Appendix D.

### 5.3. The Final Regret Bound

**Theorem 5.6.** *Fix $\delta > 0$. For any episodic MDP with trajectory feedback, with probability $1 - \delta$, the regret in $K$ episodes of Algorithm 1 is upper bounded by*

$$\text{Regret}(K) \leq \tilde{O}\big(\sqrt{SAH^3K} + \sqrt{S^3A^2H^3}K^{\frac{3}{8}} + \sqrt{S^{11}A^3H^{19}}K^{\frac{1}{4}} + \sqrt{S^{17}A^3H^{27}}\big).$$

Below we sketch the proof of Theorem 5.6.

**Regret in the first stage.** Recall the definition of $\Pi_1$ in Line 3 Algorithm 1. By Lemma 5.2 and the fact that $\pi^* \in \Pi_{\text{det}}$ with probability $1 - \delta$, we have that

- $\pi^* \in \Pi_1$;

- $\tilde{P}$ is a $(3, \sigma_0)$-approximation of $P$ with respect to $\Pi_1$, hence it is also a $(3, \sigma_0)$-approximation of $P$ with respect to $\Pi_\ell$ for any $\ell \geq 1$;

- $W^\pi(R, P) \geq W^*(R, P) - 2\epsilon_0$ for all $\pi \in \Pi_1$.

By the third property, the regret in the first stage is at most

$$O\big(\bar{K}_2\epsilon_0 + \bar{K}_1 H\big)$$
$$= \tilde{O}\left(\sqrt{SAH^3K} + S^{\frac{11}{2}}A^{\frac{3}{2}}H^{\frac{19}{2}}K^{\frac{1}{4}} + S^{\frac{17}{2}}A^{\frac{3}{2}}H^{\frac{27}{2}}\right), \tag{12}$$

where $\bar{K}_1 = 1000\sqrt{SAHK}$ and $\bar{K}_2 = K_0 - 3\bar{K}_1$ are defined in Algorithm 2 and Appendix A.

**Regret in the second stage.** Recall that the second stage comprises consecutive batches. We bound the regret in each batch separately.

Fix $1 \leq \ell \leq L = O(\log\log(K))$ and assume $\pi^* \in \Pi_\ell$. Recall that $\iota = \log^2\big(\frac{16SAK}{\delta}\big)$. Set $\tilde{x} = \sigma_0$, $\tilde{y} = 2\epsilon_\ell$, and $\epsilon_\ell = 20\big(72\sqrt{\frac{SAH^3\iota}{K_\ell}} + 9\sqrt{\frac{S^2AH^2\epsilon_0\iota}{K_\ell}} + \frac{100S^2AH^3\iota}{K_\ell} + SAH^2\sigma_0\iota\big)$. We can then verify the conditions in Lemma 5.5:

- $\pi^* \in \Pi_\ell$;

- $\tilde{P}$ is a $(3, \tilde{x})$-approximation of $P$ with respect to $\Pi_\ell$;

- $W^\pi(R, P) \geq W^*(R, P) - \tilde{y}$ for any $\pi \in \Pi_\ell$;

- $\epsilon_\ell \geq 20\big(72\sqrt{\frac{SAH^3\iota}{K_\ell}} + 6\sqrt{\frac{S^2AH^2\tilde{y}\iota}{K_\ell}} + \frac{100S^2AH^3\iota}{K_\ell} + SAH^2\tilde{x}\iota\big)$.

Using Lemma 5.5, with probability $1 - \delta$, it holds that: (1) $\pi^* \in \Pi_{\ell+1}$; (2) $W^\pi(R, P) \geq W^*(R, P) - 2\epsilon_\ell$ for any $\pi \in \Pi_{\ell+1}$. By induction on $\ell = 1, 2, \ldots, L$, with probability $1 - \frac{L\delta}{(L+1)}$, it holds that

$$W^\pi(R, P) \geq W^*(R, P) - 2\epsilon_\ell.$$

Recall that $K_\ell = 2K^{1-\frac{1}{2^\ell}}$ for $1 \leq \ell \leq L - 1$ and $K_L \leq 2K^{1-\frac{1}{2^L}}$. The regret in the $(\ell + 1)$-th batch is bounded by

$$O(K_{\ell+1}\epsilon_\ell) = \tilde{O}\big(\sqrt{SAH^3K} + \sqrt{S^3A^2H^3}K^{\frac{3}{8}} + \sqrt{S^6A^2H^7}K^{\frac{1}{4}} + S^2AH^3K^{\frac{1}{2^{\ell+1}}}\big) \tag{13}$$

for $1 \leq \ell \leq L - 1$. Moreover, the regret incurred in the first batch is bounded by $O(K_1H) = O(\sqrt{KH^2})$.

**Putting all together.** By (12) and (13), we obtain that the total regret is bounded by

$$\text{Regret}(K) \leq O\left(\bar{K}_2\epsilon_0 + \bar{K}_1 + \sum_{\ell=2}^{L} K_\ell\epsilon_{\ell-1} + K_1H\right)$$
$$= \tilde{O}\big(\sqrt{SAH^3K} + \sqrt{S^3A^2H^3}K^{\frac{3}{8}} + \sqrt{S^{11}A^3H^{19}}K^{\frac{1}{4}} + \sqrt{S^{17}A^3H^{27}}\big).$$

The proof is finished by replacing $\delta$ with $\frac{\delta}{16S^2AH(L+1)}$.

## 6. Conclusion

In this work, we design an algorithm to achieve an asymptotically nearly optimal regret bound of $\tilde{O}(\sqrt{SAH^3K})$ for RL with trajectory feedback. However, the proposed algorithm is based on elimination, resulting in exponential running time. An intriguing future direction to investigate whether the optimal regret bound is achievable using a computationally efficient algorithm. Additionally, another interesting future direction involves minimizing the lower-order terms in the regret bound.

## Acknowledgments

Y. Chen is supported in part by the Sloan Research Fellowship, the Google Research Scholar Award, the AFOSR

grant FA9550-22-1-0198, the ONR grant N00014-22-1-2354, and the NSF grants CCF-2221009 and CCF-1907661. JDL acknowledges support of Open Philanthropy, NSF IIS-2107304, NSF CCF-2212262, ONR Young Investigator Award, NSF CAREER Award 2144994, and NSF CCF-2019844. SSD acknowledges the support of NSF IIS-2110170, NSF DMS-2134106, NSF CCF-2212261, NSF IIS-2143493, NSF CCF-2019844, NSF IIS-2229881, and the Sloan Research Fellowship.

## Impact Statement

This paper discusses research aimed at advancing the field of theoretical reinforcement learning. While our work may have various potential societal impacts, we believe none of them require specific emphasis in this context.

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

## A. Parameter Settings

Set $K_0 = 100000 S^{\frac{9}{2}} A^{\frac{3}{2}} H^{\frac{17}{2}} K^{\frac{1}{2}} \log\left(\frac{SAHK}{\delta}\right)$ and $K_\ell = 2K^{1-\frac{1}{2^\ell}}$ for $\ell \geq 1$. Let $L := \min_{\ell'}(K_0 + \sum_{\ell=1}^{\ell'} K_\ell) \geq K$. Set $\epsilon_0 = 90000 \log^3\left(\frac{SAHK}{\delta}\right)\left(\frac{SAH^2}{K^{\frac{1}{4}}} + \frac{S^4 AH^6}{K^{\frac{1}{2}}}\right)$, $\sigma_0 = \frac{1}{S^{\frac{3}{2}} A^{\frac{1}{2}} H^{\frac{7}{2}} K^{\frac{1}{2}}}$, $\iota = \log^2\left(\frac{16SAHT}{\delta}\right)$ and $\kappa = 20\left(72\sqrt{\frac{SAH^3\iota}{T}} + 9\sqrt{\frac{S^2 AH^2 \epsilon_0 \iota}{T}} + \frac{100 S^2 AH^3 \iota}{T} + SAH^2 \sigma_0 \iota\right)$.

By this definition, we have $L \leq 2\log_2 \log(K)$. With a slightly abuse of notation, we re-define $K_L = K - (K_0 + \sum_{\ell=1}^{L-1} L_\ell)$. It then holds that $K_0 + \sum_{\ell=1}^{L} K_\ell = K$.

## B. Technical Lemmas

**Lemma B.1** (General Equivalence Theorem in (Kiefer & Wolfowitz, 1960)). *For any bounded subset $X \subset \mathbb{R}^d$, there exists a distribution $\mathcal{K}(X)$ supported on $X$, such that for any $\epsilon > 0$, it holds that*

$$\max_{x \in X} x^\top \left(\epsilon \mathbf{I} + \mathbb{E}_{y \sim \mathcal{K}(X)}[yy^\top]\right)^{-1} x \leq d. \tag{14}$$

*Furthermore, there exists a mapping $\pi^{\mathsf{G}}$, which maps a context $X$ to a distribution over $X$ such that*

$$\max_{x \in X} x^\top (\epsilon \mathbf{I} + \mathbb{E}_{y \sim \pi^{\mathsf{G}}(X)}[yy^\top])^{-1} x \leq 2d. \tag{15}$$

*In particular, when $\mathrm{supp}(X)$ has a finite size, $\pi^{\mathsf{G}}(X)$ could be implemented within $\mathrm{poly}(|\mathrm{supp}(X)|)$ time.*

**Lemma B.2** (Generalized KW Theorem). *For any policy set $\Pi \subset \Pi_{\mathrm{det}}$,*

$$\min_{\bar{\pi} \in \Delta^\Pi} \max_{\pi \in \Pi} \sum_{\tau \in \mathcal{T}} \mathrm{Pr}_{\pi, P}[\tau] \phi_\tau^\top (\Lambda(\bar{\pi}))^{-1} \phi_\tau = SAH, \tag{16}$$

*where $\Lambda(\pi) := \sum_{\tau \in \mathcal{T}} \mathrm{Pr}_{\pi, P}[\tau] \phi_\tau \phi_\tau^\top$.*

*Proof of Lemma B.2.* Let $F(\pi) := \log(\det(\Lambda(\pi)))$ for $\pi \in \Delta^\Pi$. Because $\Delta^\Pi$ is a closed set and $F(\pi) \leq d\log(d)$ for any $\pi \in \Delta^\Pi$ with $d = SAH$, there exists some $\bar{\pi}$ such that $\bar{\pi} = \arg\max_{\pi \in \Delta^\Pi} F(\pi)$. We assume that $\det(\Lambda(\bar{\pi})) \neq 0$. Otherwise $\det(\Lambda(\pi))$ is always 0, which implies there is redundant dimension.

Because $\bar{\pi}$ could be viewed as a distribution over $\Pi$, we use $\lambda(\bar{\pi}, \pi)$ to denote the probability that $\bar{\pi}$ distributes on $\pi$ for $\pi \in \Pi$. It then holds that $\sum_{\pi \in \Pi} \lambda(\bar{\pi}, \pi) = 1$ and

$$\Lambda(\bar{\pi}) = \sum_{\pi \in \Pi} \lambda(\bar{\pi}, \pi) \Lambda(\pi).$$

As a result, $F(\bar{\pi})$ could be viewed as a multi-variable function with respect to the distribution $\{\lambda(\bar{\pi}, \pi)\}_{\pi \in \Pi}$. For two different $\pi_1, \pi_2 \in \Pi$ such that $\lambda(\bar{\pi}, \pi_1) > 0, \lambda(\bar{\pi}, \pi_2) > 0$, by the condition that $\bar{\pi} = \arg\max_{\pi \in \Delta^\Pi} F(\pi)$, we have that

$$\frac{\partial F(\bar{\pi})}{\partial \lambda(\bar{\pi}, \pi_1)} = \frac{\partial F(\bar{\pi})}{\partial \lambda(\bar{\pi}, \pi_2)}, \tag{17}$$

which means that

$$\sum_{\tau \in \mathcal{T}} \mathrm{Pr}_{\pi_1, P}[\tau] \phi_\tau^\top (\Lambda(\bar{\pi}))^{-1} \phi_\tau = \sum_{\tau \in \mathcal{T}} \mathrm{Pr}_{\pi_2, P}[\tau] \phi_\tau^\top (\Lambda(\bar{\pi}))^{-1} \phi_\tau.$$

For $\pi_1, \pi_2$ such that $\lambda(\bar{\pi}, \pi_1) > 0$ and $\lambda(\bar{\pi}, \pi_2) = 0$, we have that

$$\frac{\partial F(\bar{\pi})}{\partial \lambda(\bar{\pi}, \pi_1)} \geq \frac{\partial F(\bar{\pi})}{\partial \lambda(\bar{\pi}, \pi_2)},$$

which implies

$$\sum_{\tau \in \mathcal{T}} \Pr_{\pi_1, P}[\tau] \phi_\tau^\top (\Lambda(\bar{\pi}))^{-1} \phi_\tau \geq \sum_{\tau \in \mathcal{T}} \Pr_{\pi_2, P}[\tau] \phi_\tau^\top (\Lambda(\bar{\pi}))^{-1} \phi_\tau.$$

Therefore, $\max_{\pi \in \Pi} \sum_{\tau \in \mathcal{T}} \Pr_{\pi, P}[\tau] \phi_\tau^\top (\Lambda(\bar{\pi}))^{-1} \phi_\tau$ is reached by any $\pi$ such that $\lambda(\bar{\pi}, \pi) > 0$. Assume this value is $x$. That is,

$$\lambda(\bar{\pi}, \pi) \sum_{\tau \in \mathcal{T}} \Pr_{\pi, P}[\tau] \phi_\tau^\top (\Lambda(\bar{\pi}))^{-1} \phi_\tau = \lambda(\bar{\pi}, \pi) x$$

for all $\pi \in \Pi$. Taking sum over $\pi \in \Pi$, we have that

$$x = \text{Trace}(\Lambda(\bar{\pi})(\Lambda(\bar{\pi}))^{-1}) = d = SAH. \tag{18}$$

The proof is completed. $\qquad\qquad\qquad\qquad\qquad\qquad\qquad\qquad\qquad\qquad\qquad\qquad\qquad\qquad\qquad\qquad$ $\square$

**Lemma B.3** (Lemma 1 in (Zhang et al., 2022b)). *Let $d > 0$ be an integer. Let $\mathcal{X} \subset (\Delta^d)^m$. Then there exists a distribution $\mathcal{D}$ over $\mathcal{X}$, such that*

$$\max_{x = \{x_i\}_{i=1}^{dm} \in \mathcal{X}} \sum_{i=1}^{dm} \frac{x_i}{y_i} = md,$$

*where $y = \{y_i\}_{i=1}^{dm} = \mathbb{E}_{x \sim \mathcal{D}}[x]$.*

**Lemma B.4** (Bennet's inequality). *Let $Z, Z_1, ..., Z_n$ be i.i.d. random variables with values in $[0,1]$ and let $\delta > 0$. Define $\mathbb{V}Z = \mathbb{E}\left[(Z - \mathbb{E}Z)^2\right]$. Then we have*

$$\mathbb{P}\left[\left|\mathbb{E}[Z] - \frac{1}{n}\sum_{i=1}^n Z_i\right| > \sqrt{\frac{2\mathbb{V}Z \ln(2/\delta)}{n}} + \frac{\ln(2/\delta)}{n}\right] \leq \delta.$$

**Lemma B.5** (Theorem 4 in (Maurer & Pontil, 2009)). *Let $Z, Z_1, ..., Z_n$ ($n \geq 2$) be i.i.d. random variables with values in $[0,1]$ and let $\delta > 0$. Define $\bar{Z} = \frac{1}{n}\sum_{i=1}^n Z_i$ and $\hat{V}_n = \frac{1}{n}\sum_{i=1}^n (Z_i - \bar{Z})^2$. Then we have*

$$\mathbb{P}\left[\left|\mathbb{E}[Z] - \frac{1}{n}\sum_{i=1}^n Z_i\right| > \sqrt{\frac{2\hat{V}_n \ln(2/\delta)}{n-1}} + \frac{7\ln(2/\delta)}{3(n-1)}\right] \leq \delta.$$

**Lemma B.6** (Lemma 10 in (Zhang et al., 2022a)). *Let $X_1, X_2, \ldots$ be a sequence of random variables taking value in $[0, l]$. For any $k \geq 1$, let $\mathcal{F}_k$ be the $\sigma$-algebra generated by $(X_1, X_2, \ldots, X_k)$, and define $Y_k := \mathbb{E}[X_k \mid \mathcal{F}_{k-1}]$. Then for any $\delta > 0$, we have*

$$\mathbb{P}\left[\exists n, \sum_{k=1}^n X_k \geq 3\sum_{k=1}^n Y_k + l\log\frac{1}{\delta}\right] \leq \delta$$

$$\mathbb{P}\left[\exists n, \sum_{k=1}^n Y_k \geq 3\sum_{k=1}^n X_k + l\log\frac{1}{\delta}\right] \leq \delta.$$

**Lemma B.7.** *Fix $d > 0$. Let $\Lambda \in \mathbb{R}^{d \times d}$ be a PSD matrix and $x \in \mathbb{R}^d$ be a vector such that $x^\top \Lambda^{-1} x \leq 1$. Then we have that*

$$\log(\det(\Lambda + xx^\top)) - \log(\det(\Lambda)) \geq 2x^\top \Lambda^{-1} x.$$

*Proof.* Direct computation gives that

$$\log(\det(\Lambda + xx^\top)) - \log(\det(\Lambda)) = \log(\det(\mathbf{I} + x^\top \Lambda^{-1} x^\top)) = \log(1 + x^\top \Lambda^{-1} x) \geq \frac{1}{2}x^\top \Lambda^{-1} x.$$

$\qquad\qquad\qquad\qquad\qquad\qquad\qquad\qquad\qquad\qquad\qquad\qquad\qquad\qquad\qquad\qquad\qquad\qquad\qquad\qquad\qquad$ $\square$

**Lemma B.8** (Proposition 1 in (Zhang et al., 2021)). *Consider a sequence of independent PSD (positive semi-definite) matrices $X_1, X_2, \ldots, X_n \in \mathbb{R}^{d \times d}$ such that $X_k \preccurlyeq W$ for a fixed PSD matrix $W$ and all $1 \leq k \leq n$. For every $\delta > 0$ and $\epsilon \in (0, 1)$, it holds that*

$$\Pr\left[\sum_{k=1}^{n} X_k \preccurlyeq 3 \sum_{k=1}^{n} \mathbb{E}[X_k] + 3 \log(d/\delta) W\right] \geq 1 - \delta; \tag{19}$$

$$\Pr\left[\sum_{k=1}^{n} X_k \succcurlyeq \frac{1}{3} \sum_{k=1}^{n} \mathbb{E}[X_k] - 3 \log(d/\delta) W\right] \geq 1 - \delta. \tag{20}$$

**Lemma B.9.** *Assume $p$ is an $(n, x)$-approximation of $p'$ with respec to $\Pi$. It then holds that*

$$\frac{1}{n} \mathbb{E}_{\pi, p}[\mathbb{I}[(s_h, a_h) = (s, a)]] \leq \mathbb{E}_{\pi, p'}[\mathbb{I}[(s_h, a_h) = (s, a)]] \leq n \mathbb{E}_{\pi, p}[\mathbb{I}[(s_h, a_h) = (s, a)]] + x \tag{21}$$

*for any $\pi \in \Pi$ and $(s, a, h)$.*

*Proof.* By (9) and (10), for any trajectory $\tau$, we have that $\frac{1}{n} \Pr_p[\tau] \leq \Pr_{p'}[\tau']$. It then holds that

$$\frac{1}{n} \mathbb{E}_{\pi, p}[\mathbb{I}[(s_h, a_h) = (s, a)]] \leq \mathbb{E}_{\pi, p'}[\mathbb{I}[(s_h, a_h) = (s, a)]].$$

On the other hand,

$$\begin{aligned}
&\mathbb{E}_{\pi, p'}[\mathbb{I}[(s_h, a_h) = (s, a)]] \\
&\leq \mathbb{E}_{\pi, p'}[\mathbb{I}[(s_h, a_h) = (s, a)] \cap \mathbb{I}[(s_{h'}, a_{h'}, s_{h+1}, h') \in \mathcal{K}, \forall 1 \leq h' \leq h]] + \max_{\pi \in \Pi_{\mathrm{det}}} \Pr_{\pi, p'}[\mathcal{K}^{\mathrm{C}}] \\
&\leq n \mathbb{E}_{\pi, p}[\mathbb{I}[(s_h, a_h) = (s, a)]] + x.
\end{aligned} \tag{22}$$

$\square$

**Lemma B.10.** *Assume $p$ is an $(n, x)$-approximation of $p'$. It then holds that*

$$\max_{\pi \in \Pi_{\mathrm{det}}} \Pr_{\pi, p'}[\mathcal{T}_{\mathrm{bad}}] \leq x,$$

*where $\mathcal{T}_{\mathrm{bad}} := \{\tau : \Pr_{p'}[\tau] \geq n \Pr_p[\tau]\}$.*

*Proof.* Let $\tau = \{s_h, a_h\}_{h=1}^{H}$ be an element in $\mathcal{T}_{\mathrm{bad}}$. By definition, there exists $h$ such that $(s_h, a_h, h, s_{h+1}) \in \mathcal{K}^{\mathrm{C}}$. As a result, $\max_{\pi \in \Pi_{\mathrm{det}}} \Pr_{\pi, p'}[\mathcal{T}_{\mathrm{bad}}] \leq \max_{\pi \in \Pi_{\mathrm{det}}} \mathrm{pr}_{\pi, p'}[\mathcal{K}^{\mathrm{C}}] \leq x$. $\square$

## C. The `Raw-Exploration` Algorithm and Analysis

**Lemma C.1.** *By running `Raw-Exploration` with input $(\Pi, \check{K}, \delta)$, with probability $1 - \delta$, the output $p$ is an $\left(3, \frac{11000 S^3 A H^4 \log(SAH/\delta)}{\check{K}}\right)$-approximation of $P$ with respect to $\Pi$.*

*Proof.* Let $\mathcal{D}^h$ be the value of $\mathcal{D}$ after the $h$-th iteration. Let $\mathcal{P}^h = $ `Confidence-Region`$(\mathcal{D}^h)$ and $\bar{\mathcal{P}}$ be the final value of $\mathcal{P}$. Let $N_{h'}^h(s, a, s')$ be the count of $(s, a, h', s')$ in $\mathcal{D}_h$ and $N_{h'}^h(s, a) := \min\{\sum_{s'} N_{h'}^h(s, a, s'), 1\}$. Let $\hat{p}_{s,a,h'}^h = \frac{N_{h'}^h(s,a,s')}{N_{h'}^h(s,a)}$ be the empirical transition probability computed by $\mathcal{D}_h$.

By Lemma B.4, with probability $1 - \delta/2$,

$$\left|\hat{p}_{s,a,h',s'}^h - P_{s,a,h,s'}\right| \leq \sqrt{\frac{4 N_{h'}^h(s, a, s') \iota}{(N_{h'}^h(s, a))^2}} + \frac{5\iota}{N_h^{h'}(s, a)} \tag{23}$$

---

**Algorithm 6** `Raw-Exploration`$(\Pi, \check{K})$

---

1: **Input**: policy set $\Pi$, length $\check{K}$ ;
2: **Initialize:** $\check{K}_1 \leftarrow \frac{\check{K}}{SAH}$, $\iota \leftarrow \log\left(\frac{2S^2AH^2}{\delta}\right)$, $\mathcal{D} \leftarrow \emptyset$;
3: **for** $h = 1, 2, \ldots, H$ **do**
4:     $\mathcal{P} \leftarrow$ `Confidence-Region`$(\mathcal{D})$;
5:     **for** $(s, a) \in \mathcal{S} \times \mathcal{A}$ **do**
6:         $\{\pi^{s,a,h}, p^{s,a,h}\} \leftarrow \arg\max_{\pi \in \Pi, p \in \mathcal{P}} \mathbb{E}_{\pi,p}\left[\mathbb{I}[(s_h, a_h) = (s, a)]\right]$;
7:     **end for**
8:     **for** $(s, a, h) \in \mathcal{S} \times \mathcal{A} \times [H]$ **do**
9:         Execute $\pi^{s,a,h}$ for $\check{K}_1$ episodes, and collect the samples as $\mathcal{D}_{s,a,h}$;
10:     **end for**
11:     $\mathcal{D} \leftarrow \mathcal{D} \cup (\cup_{s,a,h}\mathcal{D}_{s,a,h})$;
12: **end for**
13: $\mathcal{P} \leftarrow$ `Confidence-Region`$(\mathcal{D})$;
14: $p \leftarrow$ arbitrary element in $\mathcal{P}$
15: **return:** $p$;

16: **Function**: `Confidence-Region`$(\mathcal{D})$:
17:     $N_h(s, a, s') \leftarrow$ count of $(s, a, h, s')$ in $\mathcal{D}$, for all $(s, a, s')$;
18:     $N_h(s, a) \leftarrow \max\{\sum_{s'} N_h(s, a, s'), 1\}$ for all $(s, a)$;
19:     $\hat{p}_{s,a,h,s'} \leftarrow \frac{N_h(s,a,s')}{N_h(s,a)}, \forall(s, a, h, s')$;
20:     $\mathcal{W} \leftarrow \{(s, a, h, s') : N_h(s, a, s') \geq 200H^2\iota\}$;
21:     $\tilde{\mathcal{P}}_{s,a,h} \leftarrow \left\{p \in \Delta^S \mid |p_{s'} - \hat{p}_{s,a,h,s'}| \leq \sqrt{\frac{4N_h(s,a,s')\iota}{N_h^2(s,a)}} + \frac{5\iota}{N_h(s,a)}, \forall s' \in \mathcal{S}\right\}, \forall(h, s, a)$;
22:     $\mathcal{P}_{s,a,h} \leftarrow \{\text{clip}(p, \mathcal{W}) : p \in \tilde{\mathcal{P}}_{s,a,h}\}, \forall(s, a, h)$;
23:     **Return**: $\otimes_{s,a,h}\mathcal{P}_{s,a,h}$.

24: **Function**: `clip`$(p, \mathcal{W})$
25:     $p'_{s,a,h,s'} \leftarrow p_{s,a,h,s'}, \forall(h, s, a, s) \in \mathcal{W}$;
26:     $p'_{s,a,h,s'} \leftarrow 0, \forall(s, a, h, s') \notin \mathcal{W}$;
27:     $p'_{s,a,h,z} \leftarrow \sum_{s':(s,a,h,s')\notin\mathcal{W}} p_{s,a,h,s'}, \forall(h, s, a) \in [H] \times \mathcal{S} \times \mathcal{A}$;
28:     $p'_{z,a,h} \leftarrow \mathbf{1}_z, \forall(h, a) \in [H] \times \mathcal{A}$;
29:     **Return**: $p$.

---

holds for all $(s, a, h', s')$ and $h \in [H]$. We proceeds the analysis conditioned on (23). Let $N_h(s, a, s')$ denote the count of $(s, a, h, s')$ in $\mathcal{D}_{h,s,a}$ and $N_h(s, a) = \max\{\sum_{s'} N_h(s, a, s'), 1\}$. Define

$$\mathcal{K}_h := \{(s, a, s') : N_h(s, a, s') \geq 200H^2\iota\}$$

where $\iota = \log\left(\frac{2S^2AH^2}{\delta}\right)$.

By (23), for any $(s, a, s') \in \mathcal{K}_h$ and any $h' \geq h$, we have that

$$\left|\hat{p}_{s,a,h,s'}^{h'} - P_{s,a,h,s'}\right| \leq \hat{p}_{s,a,h,s'}^{h'} \cdot \left(\sqrt{\frac{1}{50H^2}} + \frac{1}{40H^2}\right),$$

which implies that

$$\left|\hat{p}_{s,a,h,s'}^{h'} - P_{s,a,h,s'}\right| \leq \frac{1}{6H}P_{s,a,h,s'}. \tag{24}$$

Moreover, by definition of $\mathcal{P}^h$, using similar arguments, we have

$$|p_{s,a,h,s'} - P_{s,a,h,s'}| \leq \frac{1}{3H}P_{s,a,h,s'} \tag{25}$$

for any $(s, a, h, s') \in \mathcal{K}_h$ and $p \in \mathcal{P}^h$.

We set $\mathcal{K} = \cup_h \mathcal{K}_h$ and verify the three conditions in Definition 5.1. The first condition (9) holds by (25), and the second condition (10) holds because $p_{s,a,h,s'} = 0$ for any $p \in \mathcal{P}$ and $(s, a, s') \in \mathcal{K}_h^C$. As for the third condition (11), we analyze as below.

Fix $h \in [H]$. By (24) and definition of $\{\pi^{s,a,h+1}, p^{s,a,h+1}\}$, we have that

$$
\mathbb{E}_{\pi^{s,a,h+1},P}\left[\mathbb{I}[(s_{h+1}, a_{h+1}) = (s, a)]\right]
$$
$$
\geq \left(1 - \frac{1}{3H}\right)^H \mathbb{E}_{\pi^{s,a,h+1},p^{s,a,h+1}}\left[\mathbb{I}[(s_{h+1}, a_{h+1}) = (s, a)]\right]
$$
$$
\geq \frac{1}{3}\mathbb{E}_{\pi^{s,a,h+1},p^{s,a,h+1}}\left[\mathbb{I}[(s_{h+1}, a_{h+1}) = (s, a)]\right]
$$
$$
\geq \frac{1}{3} \max_{\pi \in \Pi} \mathbb{E}_{\pi,p^{s,a,h+1}}\left[\mathbb{I}[(s_{h+1}, a_{h+1}) = (s, a)]\right]
$$
$$
\geq \frac{1}{9} \max_{\pi \in \Pi} \mathbb{E}_{\pi,P}\left[\mathbb{I}[(s_{h'}, a_{h'}, s_{h'+1}) \in \mathcal{K}_h, \forall 1 \leq h' \leq h] \cdot \mathbb{I}[(s_{h+1}, a_{h+1}) = (s, a)]\right]. \tag{26}
$$

Here (26) holds because for any trajectory $\tau = \{s_{h'}, a_{h'}\}_{h'=1}^h$ such that $(s_{h'}, a_{h'}, s_{h'+1}) \in \mathcal{K}_{h'}$, $\Pr_{\pi,p}[\tau] \geq \frac{1}{3}\Pr_{\pi,P}[\tau]$ for any $p \in \mathcal{P}^h$ and any $\pi \in \Pi$. Consequently,

$$
\mathbb{E}_{\pi^{s,a,h+1},P}\left[\mathbb{I}[(s_{h+1}, a_{h+1}, s_{h+2}) = (s, a, s')]\right]
$$
$$
\geq \frac{1}{9} \max_{\pi,P} \max_{\pi \in \Pi} \mathbb{E}_{\pi,P}\left[\mathbb{I}[(s_{h'}, a_{h'}, s_{h'+1}) \in \mathcal{K}_h, \forall 1 \leq h' \leq h] \cdot \mathbb{I}[(s_{h+1}, a_{h+1}, s_{h+2}) = (s, a, s')]\right]. \tag{27}
$$

On the other side, by Lemma B.6, with probability $1 - \frac{\delta}{2S^2AH^2}$, it holds that

$$
N_{h+1}(s, a, s')
$$
$$
\geq \frac{1}{3}\check{K}_1 \mathbb{E}_{\pi^{s,a,h+1},P}\left[\mathbb{I}[(s_{h+1}, a_{h+1}, s_{h+2}) = (s, a, s')]\right] - \log\left(\frac{2S^2AH^2}{\delta}\right)
$$
$$
\geq \frac{1}{27}\check{K}_1 \max_{\pi \in \Pi} \mathbb{E}_{\pi,P}\left[\mathbb{I}[(s_{h'}, a_{h'}, s_{h'+1}) \in \mathcal{K}_h, \forall 1 \leq h' \leq h] \cdot \mathbb{I}[(s_{h+1}, a_{h+1}) = (s, a)]\right] - \log\left(\frac{2S^2AH^2}{\delta}\right),
$$

which implies that

$$
\max_{\pi \in \Pi} \mathbb{E}_{\pi,P}\left[\mathbb{I}[(s_{h'}, a_{h'}, s_{h'+1}) \in \mathcal{K}_h, \forall 1 \leq h' \leq h] \cdot \mathbb{I}[(s_{h+1}, a_{h+1}) = (s, a)]\right] \leq \frac{5427H^2\iota}{\check{K}_1} \tag{28}
$$

for $(s, a, s') \in \mathcal{K}_{h+1}^C$

Taking sum over all $(s, a, s') \in \mathcal{K}_{h+1}^C$, we learn that

$$
\max_{\pi \in \Pi} \mathbb{E}_{\pi,P}\left[\mathbb{I}[(s, a, s') \in \mathcal{K}_{h+1}^C] \cdot \mathbb{I}[s_{h'}, a_{h'}, s_{h'+1}) \in \mathcal{K}_h, \forall 1 \leq h' \leq h]\right] \leq \frac{5427S^2AH^2\iota}{\check{K}_1}. \tag{29}
$$

Taking sum over $h \in [H]$, we learn that

$$
\max_{\pi} \Pr_{\pi,P}[\cup_h \mathcal{K}_h^C]
$$
$$
\leq \sum_{h=1}^H \max_{\pi \in \Pi} \mathbb{E}_{\pi,P}\left[\mathbb{I}[(s, a, s') \in \mathcal{K}_{h+1}^C] \cdot \mathbb{I}[s_{h'}, a_{h'}, s_{h'+1}) \in \mathcal{K}_h, \forall 1 \leq h' \leq h]\right]
$$
$$
\leq \frac{5427S^2AH^3\iota}{\check{K}_1}.
$$

Therefore (11) holds with $x = \frac{5427S^2AH^3\iota}{\check{K}_1}$. The proof is completed by noting $\check{K}_1 = \frac{\check{K}}{SAH}$.

$\square$

# D. Missing Algorithms and Proofs

### D.1. Proof of Lemma 5.2

*Proof.* By Lemma C.1, with probability $1 - \frac{\delta}{4(L+1)}$, $\hat{P}_2$ is an $(3, \frac{11000S^3AH^3\log(4SAH(L+1)/\delta)}{\bar{K}_2})$-approximation of $P$ with respect to $\Pi_1$. By noting that

$$\bar{K}_2 \geq 96000 S^{\frac{9}{2}} A^{\frac{3}{2}} H^{\frac{15}{2}} K^{\frac{1}{2}} \log\left(\frac{SAHK}{\delta}\right)$$

and

$$\sigma_0 \geq \frac{11000S^3AH^3\log(4SAH(L+1)/\delta)}{\bar{K}_2},$$

we conclude that $\hat{P}_2$ is an $(3, \sigma_0)$-approximation of $P$ with respect to $\Pi_1$, and thus is an $(3, \sigma_0)$-approximation of $P$ with respect to $\Pi_\ell$ for any $\ell \geq 1$.

Let $b_1 := \frac{11000S^3AH^4\log\left(\frac{4SAH}{\delta}\right)}{\bar{K}_1}$. By Lemma C.1, with probability $1 - \frac{\delta}{4}$ $\hat{P}_1$ is an $(3, b_1)$-approximation of $P$ with respect to $\Pi_{\det}$. By Lemma 5.3, with probability $1 - \frac{\delta}{4}$, we learn that

$$
\max_{\pi \in \Pi_{\det}} \left| W^\pi(\hat{R}, P) - W^\pi(R, P) \right|
$$
$$
\leq H\sqrt{\log(SAH)\log(16/\delta)}\left(b_1 + 325\sqrt{\frac{SAH\log(K)\log(8SAH/\delta)}{\bar{K}_1}}\right)
$$
$$
\leq 1000\log^2\left(\frac{SAHK}{\delta}\right) \cdot \left(\frac{SAH^2}{K^{\frac{1}{4}}} + 4SAH^2 b_1\right). \tag{30}
$$

By Lemma 5.4 with parameters as:

$$
\Pi = \Pi_{\det};
$$
$$
x = b_1 = \frac{11000S^3AH^4\log\left(\frac{4SAH}{\delta}\right)}{\bar{K}_1};
$$
$$
y = H;
$$
$$
z := 1000\log^2\left(\frac{SAHK}{\delta}\right) \cdot \left(\frac{SAH^2}{K^{\frac{1}{4}}} + \frac{S^4AH^6}{K^{\frac{1}{2}}}\right)
$$
$$
b = 30\sqrt{\frac{2S^2AH^2\log\left(\frac{32SAH}{\delta}\right)}{\bar{K}_1}} + \frac{360S^2AH^3\log\left(\frac{32SAH}{\delta}\right)}{\bar{K}_1} + \frac{44000S^3AH^4\log\left(\frac{32S^2AH^2}{\delta}\right)}{\bar{K}_1};
$$
$$
\epsilon = \epsilon_0 = 90000\log^3\left(\frac{SAHK}{\delta}\right)\left(\frac{SAH^2}{K^{\frac{1}{4}}} + \frac{S^4AH^6}{K^{\frac{1}{2}}}\right) \geq 2(b+z) \tag{31}
$$

we have that: with probability $1 - \frac{\delta}{4}$, it holds that (1) $\pi^* \in \Pi_1$; (2) $W^\pi(R, P) \geq W^*(R, P) - 2\epsilon$ for any $\pi \in \Pi_1$.

The proof is finished.

$\square$

### D.2. Proof of Lemma 5.3

*Proof.* Let $d = SAH$. Fix $\pi \in \Pi$. By definition, we have that

$$\left| W^\pi(\hat{R}, P) - W^\pi(R, P) \right| \leq \sum_{\tau \in \mathcal{T}} \Pr_{\pi, P}[\tau] \cdot |\phi_\tau^\top(\hat{R} - R)|.$$

By Lemma D.2, with probability $1 - \delta/2$, it holds that $R \in \mathcal{R}$, which implies that

$$
\begin{aligned}
\left| W^\pi(\hat{R}, P) - W^\pi(R, P) \right| &\leq \sum_{\tau \in \mathcal{T}} \Pr_{\pi, P}[\tau] \cdot |\phi_\tau^\top(\hat{R} - R)| \\
&\leq \sum_{\tau \in \mathcal{T}} \Pr_{\pi, P}[\tau] \cdot \min\{8\sqrt{H^2 \log(SAH) \log(4/\delta) \phi_\tau^\top \hat{\Lambda}^{-1} \phi_\tau}, H\} \\
&\leq H\sqrt{\log(SAH)\log(4/\delta)} \sum_{\tau \in \mathcal{T}} \Pr_{\pi, P}[\tau] \min\left\{8\sqrt{\phi_\tau^\top \hat{\Lambda}^{-1}\phi_\tau}, 1\right\}
\end{aligned}
\tag{32}
$$

By Lemma D.3, with probability $1 - \delta/2$, $\hat{\Lambda} \succcurlyeq 3\tilde{\Lambda}$. Consequently, we have that

$$
\begin{aligned}
\left| W^\pi(\hat{R}, P) - W^\pi(R, P) \right| &\leq H\sqrt{\log(SAH)\log(4/\delta)} \sum_{\tau \in \mathcal{T}} \Pr_{\pi, P}[\tau] \min\left\{5\sqrt{\phi_\tau^\top \tilde{\Lambda}^{-1}\phi_\tau}, 1\right\} \\
&\leq H\sqrt{\log(SAH)\log(4/\delta)} \cdot \left( x + 3\sum_{\tau \in \mathcal{T}} \Pr_{\pi, p}[\tau] \min\left\{5\sqrt{\phi_\tau^\top \tilde{\Lambda}^{-1}\phi_\tau}, 1\right\} \right) \tag{33} \\
&\leq H\sqrt{\log(SAH)\log(4/\delta)} \cdot \left( x + 15\sqrt{\sum_{\tau \in \mathcal{T}} \Pr_{\pi, p}[\tau] \min\left\{\phi_\tau^\top \tilde{\Lambda}^{-1}\phi_\tau, 1\right\}} \right) \tag{34} \\
&\leq H\sqrt{\log(SAH)\log(4/\delta)} \cdot \left( x + 325\sqrt{\frac{SAH \log(\check{K})\log(2d/\delta)}{\check{K}}} \right). \tag{35}
\end{aligned}
$$

Here (33) holds by Lemma B.10, (34) is by Cauchy's inequality, and (35) is by Lemma D.1.

The proof is finished.

$\square$

**Lemma D.1.** *Let $\tilde{\Lambda}$ be the final value of $\Lambda$ in Algorithm 4. It then holds that*

$$
\max_{\pi \in \Pi} \sum_{\tau \in \mathcal{T}} \Pr_{\pi, p}[\tau] \min\{\phi_\tau^\top \tilde{\Lambda}^{-1}\phi_\tau, 1\} \leq \frac{432 SAH \log(\check{K})\log(2d/\delta)}{\check{K}}
\tag{36}
$$

*Proof.* Let $\check{K}_1 = \frac{\check{K}}{54 \log(2d/\delta)}$. Let $\Lambda^t$ be the value of $\Lambda$ before the $t$-th iteration. For any policy $\pi \in \Pi$, we have that

$$
\begin{aligned}
\sum_{\tau \in \mathcal{T}} \Pr_{\pi, p}[\tau] \cdot \min\{\phi_\tau^\top \tilde{\Lambda}^{-1}\phi_\tau, 1\} &\leq \frac{1}{\check{K}_1} \sum_{t=1}^{\check{K}_1} \sum_{\tau \in \mathcal{T}} \Pr_{\pi, p}[\tau] \cdot \min\{\phi_\tau^\top (\Lambda^t)^{-1}\phi_\tau, 1\} \\
&\leq \frac{1}{\check{K}_1} \sum_{t=1}^{\check{K}_1} \sum_{\tau \in \mathcal{T}} \Pr_{\pi^t, p}[\tau] \cdot \min\{\phi_\tau^\top (\Lambda^t)^{-1}\phi_\tau, 1\} \\
&\leq \frac{1}{\check{K}_1} \cdot 4 \log\left(\frac{\det(\tilde{\Lambda})}{\lambda^{SAH}}\right) \tag{37} \\
&\leq \frac{432 SAH \log(\check{K})\log(2d/\delta)}{\check{K}}.
\end{aligned}
$$

Here (37) is derived as following. Let $z_{t,\tau} = \phi_\tau \cdot \frac{1}{\sqrt{\max\{\phi_\tau^\top (\Lambda^t)^{-1}\phi_\tau, 1\}}}$. Then we have that $\Lambda^{t+1} = \Lambda^t + \sum_{\tau \in \mathcal{T}} \Pr_{\pi^t, p}[\tau] z_{t,\tau} z_{t,\tau}^\top$. Because $z_{t,\tau} z_{t,\tau}^\top \preccurlyeq \Lambda^t$, it holds that $\sum_{\tau \in \mathcal{T}} \Pr_{\pi^t, p}[\tau] z_{t,\tau} z_{t,\tau}^\top \preccurlyeq \Lambda^t$. Let $\prec$ be an order over all possible trajectories and $\Lambda(\tau) = \Lambda^t + \sum_{\tau' \prec \tau} \Pr_{\pi^t, p}[\tau'] z_{t,\tau'} z_{t,\tau'}^\top) \preccurlyeq 2\Lambda^t$.

As a result, we have that

$$
\log\left(\frac{\det(\Lambda^{t+1})}{\det(\Lambda^t)}\right)
$$

$$
= \sum_{\tau \in \mathcal{T}} \left(\log(\det(\Lambda(\tau) + \Pr_{\pi^t,p}[\tau]z_{t,\tau}z_{t,\tau}^\top)) - \log(\det(\Lambda(\tau)))\right)
$$

$$
\geq \frac{1}{2}\sum_{\tau \in \mathcal{T}} \Pr_{\pi^t,p}[\tau]z_{t,\tau}^\top(\Lambda(\tau))^{-1}z_{t,\tau} \tag{38}
$$

$$
\geq \frac{1}{4}\sum_{\tau \in \mathcal{T}} \Pr_{\pi^t,p}[\tau]z_{t,\tau}^\top(\Lambda^t)^{-1}z_{t,\tau}. \tag{39}
$$

Here (38) is by Lemma B.7.

$\square$

**Lemma D.2.** *With probability* $1 - \delta/2$, $R \in \mathcal{R}$.

*Proof.* Let $\lambda' = 18\lambda\log(2d/\delta)$. It is easy to see $R \in [0,1]^{SAH}$. It suffices to verify that

$$
|\phi_\tau^\top R - \phi_\tau^\top \bar{R}| \leq 8\sqrt{H^2\log(SAH)\log(2/\delta)\phi_\tau^\top\hat{\Lambda}^{-1}\phi_\tau}, \quad \forall\tau.
$$

Let $\tau^t = \{(s_h^t, a_h^t)\}_{h=1}^H$. Let $\zeta^t := Y_t - \sum_{h=1}^H R_h(s_h^t, a_h^t)$. Noting that $Y^t = \sum_{h=1}^H r_h(s_h^t, a_h^t)$ where each $r_h(s_h^t, a_h^t)$ are drawn according to $R_h(s_h^t, a_h^t)$ independently, we have that $\mathrm{E}[\exp(z\zeta^t)] \leq \exp(Hz^2/2)$ for any $z \geq 0$. For fixed $\tau$, we note that

$$
\left|\phi_\tau^\top \bar{R} - \phi_\tau^\top R\right| = \left|\phi_\tau^\top\hat{\Lambda}^{-1}\sum_{t=1}^{\check{K}}\phi_{\tau^t}\zeta^t - \lambda'\phi_\tau^\top\hat{\Lambda}^{-1}R\right|
$$

$$
\leq \left|\phi_\tau^\top\Lambda^{-1}\sum_{t=1}^{\check{K}}\phi_{\tau^t}\zeta^t\right| + \lambda'H\|\phi_\tau\hat{\Lambda}^{-1}\|_2
$$

$$
\leq \left|\phi_\tau^\top\Lambda^{-1}\sum_{t=1}^{\check{K}}\phi_{\tau^t}\zeta^t\right| + H\sqrt{\lambda'\phi_\tau^\top\hat{\Lambda}^{-1}\phi_\tau} \tag{40}
$$

$$
\leq 2\left|\phi_\tau^\top\Lambda^{-1}\sum_{t=1}^{\check{K}}\phi_{\tau^t}\zeta^t\right|. \tag{41}
$$

Here (40) holds by the fact that $\hat{\Lambda} - \lambda'\mathbf{I}$ is PSD and (41) is by the fact that $18\lambda\log(2d/\delta)H^2 \leq 1$.

Note that $\{\zeta^t\}_{t=1}^{\check{K}}$ does not change the distribution of $\{\phi_{\tau^t}\}_{t=1}^T$. Therefore, it holds that

$$
\Pr\left[\left|\phi_\tau^\top\Lambda^{-1}\sum_{t=1}^{\check{K}}\phi_{\tau^t}\zeta^t\right| \geq x \cdot \sqrt{\phi_\tau^\top\hat{\Lambda}^{-1}\phi_\tau}\right] \leq 2\exp\left(-\frac{x^2}{2H}\right). \tag{42}
$$

With a union bound of all possible choices of $\tau$, we learn that, with probability $1 - \delta$, for any $\tau$, it holds that

$$
\left|\phi_\tau^\top\bar{R} - \phi_\tau^\top R\right| \leq 8\sqrt{H^2\log(SAH)\log(4/\delta)\phi_\tau^\top\hat{\Lambda}^{-1}\phi_\tau}.
$$

The proof is completed. $\square$

**Lemma D.3.** *With probability* $1 - \delta/2$, *it holds that*

$$
\hat{\Lambda} \succcurlyeq 3\tilde{\Lambda}.
$$

*Proof.* Let $\Lambda^t \preccurlyeq \tilde{\Lambda}$ be the value of $\Lambda$ before the $t$-th round in line 4. Let $z_t = \phi_{\tau^t} \sqrt{\frac{1}{\max\{\phi_{\tau^t}^\top \tilde{\Lambda}^{-1} \phi_{\tau^t}, 1\}}}$. It is then easy to verify that $\tilde{\Lambda} \succcurlyeq z_t z_t^\top$. By Lemma B.10, we have $\Pr_p[\tau] \leq 3\Pr_{p'}[\tau]$ for any $\tau$. By noting that

$$
\begin{aligned}
\tilde{\Lambda} &= \sum_{t=1}^{\check{K}_1} \mathbb{E}_{\pi^t, p}\left[ \sum_{\tau \in \mathcal{T}} \Pr_{\pi^t, p} \phi_\tau \phi_\tau^\top \cdot \frac{1}{\max\{\phi_\tau^\top (\Lambda^t)^{-1} \phi_\tau, 1\}} \right] \\
&\preccurlyeq \sum_{t=1}^{\check{K}_1} \mathbb{E}_{\pi^t, p}\left[ \sum_{\tau \in \mathcal{T}} \Pr_{\pi^t, p} \phi_\tau \phi_\tau^\top \cdot \frac{1}{\max\{\phi_\tau^\top \tilde{\Lambda}^{-1} \phi_\tau, 1\}} \right] \\
&= \check{K}_1 \mathbb{E}_{\bar{\pi}, p}\left[ \sum_{\tau \in \mathcal{T}} \Pr_{\bar{\pi}, p} \phi_\tau \phi_\tau^\top \cdot \frac{1}{\max\{\phi_\tau^\top \tilde{\Lambda}^{-1} \phi_\tau, 1\}} \right],
\end{aligned}
$$

we have

$$
\begin{aligned}
18 \log(2d/\delta)\lambda \mathbf{I} &+ \mathbb{E}_{\pi^t, P}\left[ \sum_{t=1}^{\check{K}} z_t z_t^\top \right] \\
&\succcurlyeq 18\log(2d/\delta)\lambda \mathbf{I} + \frac{1}{3}\mathbb{E}_{\bar{\pi}, p}\left[ \sum_{t=1}^{\check{K}} \phi_{\tau^t} \phi_{\tau^t}^\top \cdot \frac{1}{\max\{\phi_{\tau^t}^\top \tilde{\Lambda}^{-1} \phi_{\tau^t}, 1\}} \right] \\
&\succcurlyeq 18\log(2d/\delta)\tilde{\Lambda}.
\end{aligned} \tag{43}
$$

By Lemma B.8, with probability $1 - \delta/2$,

$$
\sum_{t=1}^{\check{K}} z_t z_t^\top \succcurlyeq \frac{1}{3}\mathbb{E}\left[ \sum_{t=1}^{\check{K}} z_t z_t^\top \right] - 3\log(2d/\delta)\tilde{\Lambda} \succcurlyeq 3\log(2d/\delta)\tilde{\Lambda} - 18\lambda\log(2d/\delta)\mathbf{I},
$$

which means that

$$
\hat{\Lambda} \succcurlyeq 18\lambda \log(2d/\delta)\mathbf{I} + \sum_{t=1}^{\check{K}} z_t z_t^\top \succcurlyeq 3\tilde{\Lambda}.
$$

The proof is completed. $\qquad\square$

### D.3. Proof of Lemma 5.5

Let $\hat{R}$ be the reward function in line 3 Algorithm 3. By Lemma 5.3, with probability $1 - \delta/2$,

$$
\max_{\pi \in \Pi} \left| W^\pi(\hat{R}, P) - W^\pi(R, P) \right| \leq b_1 := H\sqrt{\log(SAH)\log(8/\delta)} \cdot \left( \tilde{x} + 325\sqrt{\frac{SAH \log(\check{K}) \log\left(\frac{4SAH}{\delta}\right)}{\check{K}}} \right).
$$

As a result, for any $\pi \in \Pi$,

$$
W^\pi(\hat{R}, P) - W^*(\hat{R}, P) \geq W^\pi(R, P) - W^*(R, P) - 2\max_{\pi \in \Pi}\left| W^\pi(\hat{R}, P) - W^\pi(R, P) \right| \geq \tilde{y} + 2b_1. \tag{44}
$$

Let $x = x_1$, $y = \tilde{y} + 2b_1$, $z = b_1$. Let $b = 30\sqrt{\frac{SAH^2(H+Sy)\log\left(\frac{16SAH}{\delta}\right)}{\check{K}}} + \frac{360S^2AH^3\log\left(\frac{16SAH}{\delta}\right)}{\check{K}} + 4SAH^2\tilde{x}$.

By Lemma 5.4 and the assumption that $\kappa \geq 2(b+z) = 2(b+b_1)$, it then holds that $\pi^* \in \Pi_{\text{next}}$ and

$$
W^\pi(R, P) \geq W^*(R, P) - 2\kappa
$$

for any $\pi \in \Pi_{\text{next}}$

### D.4. Proof of Lemma 5.4

**Additional notations.** We use $\{v_h^\pi(s)\}$ ($\{v_h^*(s)\}$) to denote the (optimal) value function under the policy $\pi$, transition $P$ and reward $u$. With a slight abuse of notation, we define $d_P^{\bar\pi}(s, a, h) = \mathbb{E}_{\bar\pi, P}\left[\mathbb{I}[(s_h, a_h) = (s, a)]\right]$. For two vector $x, y$ with the same dimension, we write the inner product $x^\top y$ as $xy$ for simplicity. Let $p \in \Delta^S$ be a probability distribution over $[S]$ and $v \in \mathbb{R}^S$, we define the variance function as $\mathbb{V}(p, v) = pv^2 - (pv)^2$. Define $c(s, a, h) = \mathbb{E}_{\bar\pi, p}\left[\mathbb{I}[(s_h, a_h) = (s, a)]\right]$ for all $(s, a, h)$.

Because $p$ is an $(3, x)-$approximation of $P$ with respect to $\Pi$, by Lemma B.9 we have that

$$\frac{1}{3}c(s, a, h) \le d_P^{\bar\pi}(s, a, h) \le 3c(s, a, h) + x. \tag{45}$$

Let $\mathcal{L} := \{(s, a, h) : c(s, a, h) \ge \max\{x, \frac{36\log(8SAH/\delta)}{\check K}\}\}$. By (45), $d_P^{\bar\pi}(s, a, h) \le 4x$ for $(s, a, h) \notin \mathcal{L}$. By noting that $\hat p_{s_h, a_h, h}$ is independent of $v_{h+1}^*$, using Bernstein's inequality, with probability $1 - \delta/8$,

$$\left|(\hat p_{s,a,h} - P_{s,a,h})v_{h+1}^*\right| \le 2\sqrt{\frac{\mathbb{V}(P_{s,a,h}, v_{h+1}^*)\log(8SAH/\delta)}{N_h(s, a)}} + \frac{H\log(8SAh/\delta)}{N_h(s, a)}, \quad \forall(s, a, h); \tag{46}$$

$$\left|\hat p_{s,a,h,s'} - P_{s,a,h,s'}\right| \le 2\sqrt{\frac{P_{s,a,h,s'}\log(8SAH/\delta)}{N_h(s, a)}} + \frac{H\log(8SAH/\delta)}{N_h(s, a)}, \quad \forall(s, a, h, s'). \tag{47}$$

We continue the analysis conditioned on (46) and (47). Fix $\pi \in \Pi$. Using policy difference lemma, and noting that $d_P^\pi(s, a, h) \le 4x$ for $(s, a, h) \notin \mathcal{L}$, we have that

$$\left|W^\pi(\hat R, \hat p) - W^\pi(\hat R, P)\right| = \left|\mathbb{E}_{\pi, P}\left[\sum_{h=1}^H (\hat p_{s_h, a_h, h} - P_{s_h, a_h, h})v_{h+1}^\pi\right]\right|$$

$$\le \left|\sum_{(s,a,h)\in\mathcal{L}} d_P^{\bar\pi}(s, a, h)(\hat p_{s,a,h} - P_{s,a,h})v_{h+1}^\pi\right| + 4SAH^2\left(x + \frac{36\log(8SAH/\delta)}{T}\right). \tag{48}$$

Let $F = 4SAH^2\left(x + \frac{36\log(8SAH/\delta)}{\check K}\right)$. By definition of $\mathcal{E}_1$, we further have that,

$$\left|W^\pi(\hat R, \hat p) - W^\pi(\hat R, P)\right| \tag{49}$$

$$\le \left|\sum_{(s,a,h)\in\mathcal{L}} d_P^\pi(s, a, h)(\hat p_{s,a,h} - P_{s,a,h})v_{h+1}^*\right| + \left|\sum_{(s,a,h)\in\mathcal{L}} d_P^\pi(s, a, h)(\hat p_{s,a,h} - P_{s,a,h})(v_{h+1}^\pi - v_{h+1}^*)\right| + F$$

$$\le \left|\sum_{(s,a,h)\in\mathcal{L}} d_P^\pi(s, a, h)\left(2\sqrt{\frac{\mathbb{V}(P_{s,a,h}, v_{h+1}^*)\log\left(\frac{8SAH}{\delta}\right)}{N_h(s, a)}} + \frac{H\log\left(\frac{8SAH}{\delta}\right)}{N_h(s, a)}\right)\right|$$

$$+ \left|\sum_{(s,a,h)\in\mathcal{L}} d_P^\pi(s, a, h)\left(2\sqrt{\frac{S\mathbb{V}(P_{s,a,h}, v_{h+1}^* - v_{h+1}^\pi)\log\left(\frac{8SAH}{\delta}\right)}{N_h(s, a)}} + \frac{SH\log\left(\frac{8SAH}{\delta}\right)}{N_h(s, a)}\right)\right| + F$$

$$\le 2\sqrt{\log\left(\frac{8SAH}{\delta}\right)\left(\sum_{(s,a,h)\in\mathcal{L}} \frac{d_P^\pi(s, a, h)}{N_h(s, a)}\right)} \cdot \sqrt{\mathbb{E}\left[\sum_{h=1}^H \left(\mathbb{V}(P_{s_h, a_h, h}, v_{h+1}^*) + S\mathbb{V}(P_{s_h, a_h, h}, v_{h+1}^* - v_{h+1}^\pi)\right)\right]}$$

$$+ 2SH\log\left(\frac{8SAH}{\delta}\right)\left(\sum_{(s,a,h)\in\mathcal{L}} \frac{d_P^\pi(s, a, h)}{N_h(s, a)}\right) + F. \tag{50}$$

We then bound the terms in (50) separately.

**The doubling count term.** By definition of $\mathcal{L}$, we have that

$$\sum_{(s,a,h)\in\mathcal{L}} \frac{d_P^\pi(s,a,h)}{N_h(s,a)} \leq 4 \sum_{s,a,h} \frac{d_p^\pi(s,a,h)}{N_h(s,a)}. \tag{51}$$

By Lemma B.6, we further have that, with probability $1 - \frac{\delta}{8}$, it holds that

$$N_h(s,a) \geq \frac{1}{9}\check{K}c(s,a,h) - \log(8SAH/\delta). \tag{52}$$

for any $(s,a,h)$. Conditioned on this event, we have that

$$\sum_{(s,a,h)\in\mathcal{L}} \frac{d_P^\pi(s,a,h)}{N_h(s,a)} \leq \frac{108}{\check{K}} \sum_{s,a,h} \frac{d_p^\pi(s,a,h)}{c(s,a,h)} \leq \frac{108SAH}{\check{K}}. \tag{53}$$

In the last inequality, we use the fact that

$$\max_{\pi^*\in\Pi} \sum_{s,a,h} \frac{d_p^{\pi^*}(s,a,h)}{c(s,a,h)} = SAH, \tag{54}$$

which is a direct result following Lemma B.3.

**The variance terms.** Direct computation gives that

$$
\begin{aligned}
\mathbb{E}_{\pi,P}\left[\sum_{h=1}^{H} \mathbb{V}(P_{s_h,a_h,h}, v_{h+1}^*)\right] &= \mathbb{E}_{\pi,P}\left[\sum_{h=1}^{H}\left((v_{h+1}^*(s_{h+1}))^2 - (P_{s_h,a_h,h}v_{h+1}^*)^2\right)\right] \\
&\leq \mathbb{E}_{\pi,P}\left[\sum_{h=1}^{H}\left((v_h^*(s_h))^2 - (P_{s_h,a_h,h}v_{h+1}^*)^2\right)\right] \\
&\leq 2H\mathbb{E}_{\pi,P}\left[\sum_{h=1}^{H}\left(v_h^*(s_h) - P_{s_h,a_h,h}v_{h+1}^*\right)\right] \\
&= 2H\mathbb{E}_{\pi,P}\left[\sum_{h=1}^{H}\left(v_h^*(s_h) - v_{h+1}^*(s_{h+1})\right)\right] \\
&\leq 2H^2
\end{aligned}
\tag{55}
$$

and

$$\mathbb{E}_{\pi,\mathbb{P}}\left[\sum_{h=1}^{H}\mathbb{V}(P_{s_h,a_h,h},v_{h+1}^*) + S\mathbb{V}(P_{s_h,a_h,h},v_{h+1}^* - v_{h+1}^\pi)\right]$$

$$= \mathbb{E}_{\pi,P}\left[\sum_{h=1}^{H}\left((v_{h+1}^*(s_{h+1}) - v_{h+1}^\pi(s_{h+1}))^2 - (P_{s_h,a_h,h}(v_{h+1}^* - v_{h+1}^\pi))^2\right)\right]$$

$$\leq \mathbb{E}_{\pi,P}\left[\sum_{h=1}^{H}\left((v_h^*(s_h) - v_h^\pi(s_h))^2 - (P_{s_h,a_h,h}(v_{h+1}^* - v_{h+1}^\pi))^2\right)\right]$$

$$\leq H\mathbb{E}_{\pi,P}\left[\sum_{h=1}^{H}\left|(v_h^*(s_h) - P_{s_h,a_h,h}v_{h+1}^*) - (v_h^\pi(s_h) - P_{s_h,a_h,h}v_{h+1}^\pi)\right|\right]$$

$$= 2H\mathbb{E}_{\pi,P}\left[\sum_{h=1}^{H}\left|(v_h^*(s_h) - P_{s_h,a_h,h}v_{h+1}^*) - \hat{R}_h(s_h,a_h)\right|\right]$$

$$= 2H\mathbb{E}_{\pi,P}\left[\sum_{h=1}^{H}\left(v_h^*(s_h) - u_h(s_h,a_h) - P_{s_h,a_h,h}v_{h+1}^*\right)\right] \tag{56}$$

$$2H(W^*(u,P) - W^\pi(u,P))$$

$$\leq 2Hy. \tag{57}$$

Here (56) holds by the fact that $v_h^*(s_h) \geq u_h(s_h,a_h) + P_{s_h,a_h,h}v_{h+1}^*$.

**Putting together.** By (50), (53) (55) and (57), we have that

$$|W^\pi(u,\hat{p}) - W^\pi(u,P)| \leq 30\sqrt{\frac{SAH^2(H+Sy)\log\left(\frac{8SAH}{\delta}\right)}{\check{K}}} + \frac{360S^2AH^3\log\left(\frac{8SAH}{\delta}\right)}{\check{K}} + 4SAH^2x = b. \tag{58}$$

Now we verify that $\pi^* \in \Pi_{\text{next}}$.

It suffices to show that

$$W^{\pi^*}(u,\hat{p}) \geq \max_{\pi'\in\Pi}W^{\pi'}(u,\hat{p}) - \epsilon. \tag{59}$$

By the assumptions and (58), we have that

$$W^{\pi^*}(u,\hat{p}) \geq W^{\pi^*}(u,P) - b \geq W^{\pi^*}(R,P) - b - z$$
$$W^\pi(\mu,\hat{p}) \leq W^\pi(u,P) + b \leq W^\pi(R,P) + b + z \leq W^{\pi^*}(R,P) + b + z.$$

Noting that $\epsilon \geq 2(b+z)$, we conclude that $\pi^* \in \Pi_{\text{next}}$. On the other hand, for any $\pi \in \Pi_{\text{next}}$, we have that

$$W^\pi(R,P) \geq W^\pi(u,\hat{p}) - (b+z) \geq W^{\pi^*}(u,\hat{p}) - 2(b+z) \geq W^{\pi^*}(R,P) - 3(b+z) \geq W^{\pi^*}(R,P) - 2\epsilon.$$

The proof is finished.

