# OpenReview forum: "Minimax Optimal Regret Bound for Reinforcement Learning with Trajectory Feedback"
_ICML.cc/2025/Conference — ICML 2025 poster_

### Official Review · Reviewer_zDaM · 2025-03-06

**Overall Recommendation:** 3

**Summary:**

This paper studies the setting of tabular MDP, with trajectory feedback, i.e., every round after executing a policy the learner obtains the entire trajectory with the total reward along the trajectory. The per-step reward is not given to the learner. This paper provided an algorithm for this setting and also proved that the algorithm achieves $\tilde{O}(\sqrt{SAH^3K})$ regret upper bound, which also matches the lower bound, asymptotically.

**Claims And Evidence:**

Yes

**Essential References Not Discussed:**

No. As far as I know, this paper includes sall related references.

**Experimental Designs Or Analyses:**

No experiment is provided in this paper.

**Methods And Evaluation Criteria:**

Yes

**Other Comments Or Suggestions:**

Typos:
(1) in Eq. (1), (Y^t - \phi_{\tau^t}^T r) to (Y^t - \phi_{\tau^t}^T r)^2
(2) On the right side of Line 175, d^\pi_P to d^\pi_P(\tau)
(3) In Line 722 (Line 6 of Algorithm Raw-Exploration), missing symbol "\leftarrow"

Suggestions:
1. The author mentioned that the preference-based learning can be a possible application to the RL with trajectory feedback. It would be better write down explicitly how does results in this paper implies about preference-based learning.
2.

**Other Strengths And Weaknesses:**

Strength:
1. This paper is well-written, except few typos I pointed out below.
2. The paper provides a tight characterization to the regret.

Weakness:
1. This paper studied the setting of tabular MDP, which is quite restrictive. Additionally, this paper merely tightens the bounds provided in Efroni et al. 21, which is incremental.
2. I concern about the novelty of this paper. The results are good results, and they match the lower bound, which is also good, but by looking at the proving techniques, it feels that everything is well developed in previous literature and nothing feels special to me.
3. The upper bound of the algorithm requires a large "burning in" phase, which I believe, still having improvement space.

**Questions For Authors:**

I have the following questions:
1. Regarding the "burning in" term in the regret, e.g. the term that does not scale with K, what is the best you can obtain?
2. Is it possible to generalize the results into linear MDP, or MDP with function approximation, since the tabular MDP is most restrictive?
3. The algorithm designed in this paper first learns the transition model $P$, then learn the reward model $r$. Is it possible to learn these two simultaneously?

**Relation To Broader Scientific Literature:**

The closest literature to this paper is (Efroni et al., 2021), which proposes the setting of learning MDP with trajectory rewards. In the previous paper, they show the regret upper bounded by $O(\sqrt{S^2A^2H^4K})$, which is suboptimal. This paper improve this rate to $O(\sqrt{SAH^3K})$.

**Theoretical Claims:**

Yes, I checked all proofs. They all look correct to me.

---

> ### Author Rebuttal · Authors · 2025-04-01
>
> We are grateful for the reviewer's detailed comments and constructive suggestions. Below we present our response.
>
> **Regarding significance:**  We would like  to stress that even for RL in the standard setting, minimax optimal regret bounds have not been obtained until recent few years, and the burn-in terms were not fully optimized until very recently. Therefore, as the first work that achieves minimax optimal regret bound for RL with trajectory, we believe our work is non-trivial in terms of technique.
>
> **Regarding technical novelty:** While the algorithmic framework depends on previous work, the key technical observation is that the number of possible trajectories $O( (SA^H)$ is smaller to that of possible policies $O( A^{SH})$. This is why we can remove the extra $\sqrt{S}$ factor compared to previous work Efroni et al., 21.
>
> **Regarding the large burn-in terms:**  Indeed our current regret bound has large burn-in terms, and we would leave improving those terms as a future direction.  The current term is $S^{24}A^4H^{32}$. The main reason of such a large burn-in term is due to learning the reference model. This term can be significantly reduced under the assumption of an effective reference model.
>
> **Regarding other comments:**
>
> **About the typos:**
> Thanks for pointing out the typos. We will fix the typos accordingly in the next revision. In the second typo, we use $d_{P}^{\pi}$ to denote the feature vector with respect to $\pi$, which does not depend on the trajectory $\tau$.
>
> **About the preference-based learning:** Preference-based RL (PbRL) is another RL paradigm to deal with the lack of reward signals. We will discuss the similarities and differences between the two settings in the next revision.
>
> **Regarding the questions:**
>
> **Abut the exact burn-in term:** Following the current analysis, the final regret bound is $\tilde{O}(\sqrt{SAH^3K}+S^{24}A^4H^{32})$.
>
> **About the extensions to other settings, e.g., linear MDP and MDP with function approximation:** At the moment, we are not aware of any possible extensions to linear MDP or MDP with function approximation. We believe these problem settings should enjoy some advantages of low-rank structure so that we can construct a feature space with lower dimensions.
> For example, in linear MDP, we can take $ (\phi_{s_1,a_1}, \phi_{s_2,a_2},..., \phi_{s_H,a_H})$  as a $dH$-dimensional unified feature to learn the reward kernel $ (\theta_{1}, \theta_2,..., \theta_H)$. However, the exploration and policy design over this feature space is more complicated compared to the counterparts in the tabular case. It requires more efforts to solve the experimental design problem over these complicated structures.
>
> **About learning the reward and transition kernel simultaneously:**  Thanks for the interesting question. Although we cannot establish this definitively, we conjecture the answer is negative in the worst case. The problem could be reformulated as finding a mixed policy $\bar{\pi}$ to meet two following conditions simultaneously (we set $\phi(\tau)$ to be $\phi_{\tau}$ for display):
>
> $$\min_{\bar{\pi}\in \Delta^{\Pi}}\max_{\pi\in \Pi}\sum_{\tau}Pr_{\pi,P}[\tau]\phi^{\top}(\tau)\Lambda(\bar{\pi})^{-1}\phi(\tau)=O(SAH)$$
>
> $$\min_{\bar{\pi}\in \Delta^{\Pi}}\max_{\pi\in \Pi}\sum_{s,a,h} \frac{d_{P}^{\pi}(s,a,h)}{d_{P}^{\bar{\pi}}(s,a,h)}=O(SAH).$$
>
> For the first problem, the target is to maximize $\log(\det(\Lambda(\bar{\pi})))$
> where we recall that $\Lambda(\bar{\pi}) = \sum_{\tau}Pr_{\pi,P}[\tau]\phi(\tau)\phi^{\top}(\tau)$.
>  For the second problem, the target is to maximize $\log(\Pi_{s,a,h}d_{P}^{\bar{\pi}}(s,a,h))=\log(\det(diag(\Lambda(\bar{\pi})))$, where $diag(\Lambda)$ denotes the matrix by setting all non-diagonal elements of $\Lambda$ to 0. As a result, the two optimization problem are substantially different when $h\geq 2$. So we conjecture that the answer to your question is probably negative in the worst case.

---

### Official Review · Reviewer_J7DL · 2025-03-10

**Overall Recommendation:** 3

**Summary:**

This paper investigates reinforcement learning with trajectory feedback, where the agent receives only the cumulative reward for an entire trajectory rather than individual state-action rewards, while still observing all visited state-action pairs. The authors establish the first asymptotically nearly optimal regret bound of $O(\sqrt{S A H^3 K}) $ for this setting, matching the asymptotically optimal regret bound in standard RL. To achieve this, they construct a tighter confidence region for rewards by leveraging the structure of the linear bandit instance associated with RL with trajectory feedback.

**Claims And Evidence:**

All theoretical claims are followed by proofs in the appendix or the main paper.

**Essential References Not Discussed:**

To the best of my knowledge, the related work is clearly presented.

**Experimental Designs Or Analyses:**

not applicable.

**Methods And Evaluation Criteria:**

As a theoretical paper, the proposed algorithm seems well suited for the problem.

**Other Comments Or Suggestions:**

- The phrase on page 6, lines 282-284, may be lacking some connectors, which makes the formulation unclear: ex.: "to the L1 norm"

- Page 4 line 208: $Y_t$ should be $Y^t$?

**Other Strengths And Weaknesses:**

**Strenghts**:
- The paper is generally well-written.

- The technique of building the confidence region around the reward estimation by considering trajectories, rather than defining it for each deterministic policy and applying a union bound, enables an improved dependence on the number of states in the regret bound and appears to be a novel approach.

- This work is the first to achieve an asymptotically near-optimal regret bound for this problem.

**Weaknesses**:

- Some parts of the paper require further clarification, such as the proof of Lemma B.1 and the precise application of the classical Kiefer-Wolfowitz theorem in this setting (as claimed on page 5, line 224).

- The proposed algorithm is not computationally efficient, whereas some existing approaches, such as UCBVI-TS from Efroni et al. (2021), are computationally efficient but yield worse regret bounds ($ O(H^7 S^4 A^3 K) $).

- The regret bound established in this work holds asymptotically in $ K $ and does not apply for all $ K > 0 $, unlike the bound in Efroni et al. (2021).

**Questions For Authors:**

Why the authors choose to use an elimination-based online batch learning process rather than using the new confidence region directly with a linear bandit algorithm? I may be overlooking a key step and would appreciate further clarification.

**Relation To Broader Scientific Literature:**

Efroni et al. (2021) established a regret bound of $ \tilde{O}(\sqrt{S^2 A^2 H^4 K}) $ for RL with trajectory feedback, which holds for all $ K > 0 $, not just asymptotically. The authors suggest that refining the analysis could improve this bound to $ \tilde{O}(\sqrt{S^2 A H^3 K}) $, but reducing the dependence on the number of states requires a fundamentally new approach. This paper presents the first algorithm that asymptotically achieves the lower bound of $\tilde{O}(\sqrt{S A H^3 K}) $ for RL with trajectory feedback, matching the optimal rate in standard RL, though the bound holds only in the asymptotic regime (for large $ K $, not all $ K > 0 $). Like Efroni et al. (2021), the algorithm leverages the connection between RL with trajectory feedback and linear bandits but introduces a novel confidence region around rewards. Additionally, it employs a policy elimination method to learn the transition kernel, similar to the approach of Zhang et al. (2022b).

**Theoretical Claims:**

I've reviewed some of the proofs, and while most appear correct, I would appreciate further clarification on the proof of Lemma B.1. This lemma seems to play a key role in improving the dependence of the regret bound on $S $. However, the definition of $ \lambda(\bar{\pi}, \pi) $ is unclear. How do the authors define the probability that $ \bar{\pi} $ "distributes" over $ \pi $? Additionally, the interpretation of the partial derivative of $ F $ with respect to this quantity is not well explained. These concepts may be standard in this type of analysis, but it would be helpful to reintroduce them rigorously, especially for readers in RL who may not be familiar this.

---

> ### Author Rebuttal · Authors · 2025-04-01
>
> We are grateful for the reviewer's detailed reviews and valuable suggestions. Below we present our response.
>
> **Regarding Lemma B.1:** In Lemma B.1,  $\bar{\pi}$ represents a mixed policy. That is, by following $\bar{\pi}$, the learner takes each deterministic policy $\pi\in \Pi$ with a certain probability as $\lambda(\bar{\pi},\pi)$. As a result, $F(\bar{\pi})$ could be regarded as a multi-variable function with respect to the probability vector $[\lambda(\bar{\pi},\pi)]_{\pi\in \Pi}$.  We will revise the proof accordingly in the next version.
>
> **Regarding clarification of KW theorem:** In the proof, we do not directly use KW theorem. Instead, we generalize the KW theorem to a distributed version in Lemma B.1. In the next revision, we are planning to present the original version of the KW theorem together with our generalized version to make the difference clear.
>
> **Regarding computational inefficiency:** We admit that the current algorithm is computationally inefficient, and indeed, devising algorithm with minimax optimal regret bound and polynomial running time for RL with trajectory feedback is an intriguing open problem. However, we notice that obtaining statistically (nearly) optimal algorithms is usually the first step for completely settling problems in machine learning theory, and we believe our techniques would be useful for later developments in the theory of RL with trajectory feedback.
>
> **Regarding large burn-in terms:** Indeed our current regret bound has large burn-in terms, and we would leave improving those terms as a future direction. On the other hand, we would like stress that even for RL in the standard setting, minimax optimal regret bounds have not been obtained until recent few years, and the burn-in terms were not fully optimized until very recently. Therefore, as the first work that achieves minimax optimal regret bound for RL with trajectory, we believe it is reasonable to have large burn-in terms in our bound. Moreover, we believe our new techniques are crucial for fully understanding the regret bound of RL with trajectory feedback.
>
> **Regarding other comments and suggestions:**
>
> **About lines 282-284:** We will rewrite this sentence as *"Instead of approximating $P$ by $p$ under the $L_1$-norm, it is required the trajectory distribution under $P$  could be covered by that under $p$, up to a constant ratio."*
>
> **About line 208:** Thanks for pointing out the typo. We will fix accordingly.
>
> **Regarding the  elimination-based online batch learning:** In this work, the fundamental problem is still reinforcement learning. More precisely, even if assuming that the reward is known, we are required to learn the transition kernel. This step cannot be trivially implemented using a linear bandit algorithm. As a result, we choose to use a linear bandit algorithm to learn the reward function and  an RL algorithm to learn the transition kernel.  Another two reasons to apply the eliminated-based batch learning are that: (1) Designing an algorithm that can simultaneously learn both the reward function and transition kernel presents significant challenges. As a result, we have to learn the reward and transition kernel separately; (2) By batch learning, we can efficiently reduce the statistical dependencies between different batches, which makes the analysis less complicated.

---

> > ### Comment · Reviewer_J7DL · 2025-04-04
> >
> > I thank the authors for addressing my concerns and for committing to clarify some parts of the paper (such as Lemma B.1 and the use of KW theorem). I will keep my positive score.

---

### Official Review · Reviewer_gvw8 · 2025-03-13

**Overall Recommendation:** 4

**Summary:**

This work considers the problem of online learning in a tabular finite horizon MDP with stochastic rewards and aggregate bandit feedback, where agent observes only the sum of rewards she collected after each episode.

An algorithm based on policy elimination is proposed, which builds on the linear bandits perspective to the problem. This approach, while not computationally efficient, achieves nearly minimax optimal regret in all problem parameters (assuming the number of episodes $K$ is sufficiently large).

One of the main observations used to design the algorithm, is that the number of possible trajectories in the MDP ($(SA)^H$) is much smaller than the number of "arms", i.e., deterministic policies ($A^{SH}$). Thus, building on confidence regions based on trajectories rather than policies, leads to dependence on $\sqrt S$ rather than $S$ that would be the result of a vanilla linear bandits algorithm, even in the known dynamics case.

## update after rebuttal
I chose to increase my rating to accept. After further consideration, and taking into account that none of my concerns were major, I think there is more than enough in this work to merit acceptance.

**Claims And Evidence:**

Yes

**Essential References Not Discussed:**

None that I know of.

**Experimental Designs Or Analyses:**

N/A

**Methods And Evaluation Criteria:**

N/A

**Other Comments Or Suggestions:**

Algorithms
* Algorithm 5 line 3 computes $c(s,a,h)$ (the occupancy of $\bar\pi$ wrt $p$?) but it does not seem to be used anywhere.
* Algorithm 2 does not pass a $\delta$ (lines 3,6) to Raw-Exploration (algorithm 6)
* I suggest you add a "Conditioned on parameter settings in Section A,..." to the relevant Lemma statements.
- In the proof overview of Theorem 5.6, should $\tilde y=2\epsilon_0$ be $\tilde y=2\epsilon_\ell$?

Lemma 5.2
- In the proof, $\tilde P \to \hat P_2$ (or change the Lemma statement to be about $\tilde P$)
- Eq. 28 is not synced with Lemma 5.3 (e.g., $log^2(T/\delta) \nleq \log (T) \log(1/\delta)$)
- Also in Eq. 28, the last inequality seems to follow from a condition on $K$ that is not mentioned in the statement of the Lemma.

Lemma 5.3
- “By Lemma D.2 … it holds that $R \in \mathcal R$” is this the $\mathcal R$ defined in line 13 of Algorithm 4, when invoked with $(p, T, \Pi)$ of the current Lemma? Better to be explicit about this to help the reader. Also, $\mathcal R$ denotes your reward distribution, which is unrelated.
- In Eq. 30, same comment as for Lemma 5.2, $\log^2(Z/\delta) \nleq \log (Z) \log(1/\delta)$ for general $Z$.
- There seems to be a sum over $\tau$ missing in Eqs. 31 and 32

Misc
* "To circumvent issues mentioned above, practitioners often rely on heuristics (e.g., reward shaping (Ng et al., 1999) or reward hacking (Amodei et al., 2016))." - Amodei et al. propose methods to **avoid** reward hacking. Reward hacking refers to the scenario where the agent exploits an inaccurate reward signal.
* "The optimal Q-function and V-function are given by ..." Why use $\sup$ and not $\max$ for $Q^*$?
* "we write the inner product $x^\top y$ as $xy$ for simplicity" - I would suggest to revise this decision, it generates place for confusion with scalar multiplication.
* "the regret stemming from learning $\tilde P$ can be bounded by..." It this point it is not clear what is the reference transition kernel so the sentence does not convey much information.

**Other Strengths And Weaknesses:**

**Strengths**

* The approach yields a (near) minimax optimal regret bound for the problem.
* The technical overview and presentation of the algorithm are well written and relatively clear.

**Weaknesses**

* The algorithm is not computationally efficient, and the burn-in period (requirement on $K$ for minimax optimality) is quite demanding.

Ultimately, establishing the optimal dependence on problem parameters is secondary in importance, even more so for a non-computationally efficient approach. Further, this is essentially the first paper that proposes a non-computationally efficient approach to the problem. That said, it is clear the problem is far from trivial and it seems this work offers some valuable insights. The algorithmic approach is pretty elegant and intuitive, at least at a high level.

**Questions For Authors:**

I don't have any important questions.

**Relation To Broader Scientific Literature:**

The problem we initially proposed by Efroni et al. (2021). Later, Cohen et al. (2021) study the adversarial setting, and Cassel et al. (2024) study the stochastic Linear MDP setting.

All the previous approaches provide $\widetilde O(\sqrt K)$ regret in the the tabular stochastic setting, but with suboptimal dependence on the rest of the problem parameters $S,A, H$. However, they are computationally efficient.

The approach proposed in this work gives optimal dependence on all parameters (assuming $K$ large enough), but is not computationally efficient.

**Theoretical Claims:**

I went over proofs of Lemmas 5.2 and 5.3, and the proof overview of Theorem 5.6. I didn't find any significant issues.

---

> ### Author Rebuttal · Authors · 2025-04-01
>
> We sincerely appreciate the reviewer's thorough evaluation and constructive feedback. Below we present our response.
>
> **Regarding the computational issue:**
>
> We admit that the current algorithm is computationally inefficient, and indeed, devising algorithm with minimax optimal regret bound and polynomial running time for RL with trajectory feedback is an intriguing open problem. However, we notice that obtaining statistically (nearly) optimal algorithms is usually the first step for completely settling problems in machine learning theory, and we believe our techniques would be useful for later developments in the theory of RL with trajectory feedback.
>
> **Regarding the burn-in terms:**
>
> Indeed our current regret bound has large burn-in terms, and we would leave improving those terms as a future direction. On the other hand, we would like stress that even for RL in the standard setting, minimax optimal regret bounds have not been obtained until recent few years, and the burn-in terms were not fully optimized until very recently. Therefore, as the first work that achieves minimax optimal regret bound for RL with trajectory, we believe it is reasonable to have large burn-in terms in our bound. Moreover, we believe our new techniques are crucial for fully understanding the regret bound of RL with trajectory feedback.
>
> **Regarding other comments and suggestions:**
>
> **About $c(s,a,h)$:** We are sorry for this mistake. $c(s,a,h)$ is only used  in the analysis for Algorithm 5. We will move the definition of $c(s,a,h)$ to Appendix D.4 in the next revision.
>
> **About $\delta$ in Algorithm 2:** We will use $\delta$ as a common parameter across all algorithms and delete $\delta$ from the input of Algorithm 6.
>
> **About parameters in Appendix.A:** Thanks for the suggestion. We will refine the lemma statements to improve clarity.
>
> **About the proof overview of Theorem 5.6:**  Thank you for pointing out the typo. It should be $\tilde{y}=2\epsilon_{\ell}$.
>
> **About $\tilde{P}$ in Lemma 5.2:** Thank you for pointing out the typo. We will replace $\tilde{P}\to \hat{P}_2$ accordingly.
>
> **About Eq.(28):** Here we use the fact that $\log(A)\log(B)=O(\log^2(AB))$ for $A,B\geq 1$.  We have revised the inequality and reset the value of $\epsilon_0$ as $\epsilon_0=90000\log^3(\frac{SAHK}{\delta})\left( \frac{SAH^2}{K^{\frac{1}{4}}}+\frac{S^4AH^6}{K^{\frac{1}{2}}} \right)$. There should be an extra $H$ factor in the right-hand side of Eq.(28) to make the inequality holds. The revised version of Eq.28 is as follows.
> $$\max_{\pi\in \Pi_{\mathrm{det}}}| W^{\pi}(\hat{R},P) - W^{\pi}(R,P)|\leq H\sqrt{\log(SAH)\log(16/\delta)}\left(b_1+ 325\sqrt{\frac{SAH\log(K)\log(8SAH/\delta)}{\bar{K}_1}} \right)\leq  1000\log^2\left(\frac{SAHK}{\delta}\right)\cdot \left(\frac{SAH^2}{K^{\frac{1}{4}}} + 4SAH^2b_1 \right).$$
>  We remark that this $H$ factor is in the second order term so the final regret bound is still asymptotically optimal.
>
> **About $\mathcal{R}$:** We are sorry for the abuse of notations. You are correct that $\mathcal{R}$ is the confidence region defined in line 13 of Algorithm 4. The proof of Lemma D.2 is under the proof of Lemma 5.2, which studies the properties of Algorithm 4. We will make this clear in the next revision.We will also replace $\mathcal{R}$ with $\mathsf{R}$ when introducing the reward distribution.
>
> **About Eq.(30), Eq.(31) and Eq.(32):** Thanks for pointing out the typos. We will fix accordingly.
>
> **About the term "reward hacking":** We will remove the term "reward hacking" from this paragraph.
>
> **About $\sup$:** We will replace $\sup$ with $\max$.
>
> **About $xy$ and $x^{\top}y$:** We will fix the notation about $\sup$ and $x^{\top}y$. We only use $xy$ as a shorthand of $x^{\top}y$ in Appendix D.4 to reduce the complexity of the notations.
>
> **About "the regret stemming from learning $\tilde{P}$ can be bounded by ...":** We will mention  that *"$\tilde{P}$ serves as an efficient tool to help design the exploration policy"* before this sentence.

---

### Official Review · Reviewer_egr3 · 2025-03-13

**Overall Recommendation:** 3

**Summary:**

This work studies Reinforcement Learning with only Trajectory Feedback, where the agent does not observe the reward for each individual step separately. Under this setting, the author proposes a novel algorithm based on the arm-elimination method over all possible deterministic policies and achieves a near-optimal regret bound, which matches the lower bound for the setting with single-step rewards up to logarithmic factors.

**Claims And Evidence:**

The author provides a clear claim of the result in the theorems and includes a proof sketch to outline the key steps of the theoretical analysis.

**Essential References Not Discussed:**

This paper provides a comprehensive discussion of related work in linear bandit and trajectory-based reinforcement learning method.

**Experimental Designs Or Analyses:**

The main contribution of this work focuses on the theoretical analysis of regret and does not have experiment.

**Methods And Evaluation Criteria:**

The main contribution of this work focuses on the theoretical analysis of regret and does not have experiment.

**Other Comments Or Suggestions:**

1. The notation of $K$ and $T$ is consing. As the author mention in the introduction, there may exists a gap of the episode length $H$ between the number of episode and tnumber of stage $H$. However, the author use the both notation in algorithm, e.g., Algorithm 1 take a oracle to  Traj-Learning with parameter $K$ and when introduce the  Traj-Learning algoritmh, with a notation of $T$. better to union.

2. For the non-dominant term in Theorem 5, there is still a dependency on the number of $K$. It would be better to explicitly separate the effect of $K$ in the non-dominant term to provide a clearer understanding of its impact on the regret bound.

**Other Strengths And Weaknesses:**

1. The proposed algorithm is highly computationally inefficient, making it primarily relevant for the theoretical analysis of reinforcement learning rather than practical applications.

2. For the non-dominant term, the dependency on $S, A, H$ is extremely large, making the regret guarantee non-trivial and only meaningful for an excessively large number of rounds $K$.

**Questions For Authors:**

1. As the author mentions in the regret guarantee, the upper bound matches the lower bound for a more powerful learner with single-step rewards, suggesting that the main challenge in learning an MDP comes from estimating the transition dynamics. Given this, it seems unreasonable that the algorithm in Ref-Model only dedicates a small fraction of rounds to estimating the transition matrix and then primarily focuses on reward estimation using the estimated transition model for the majority of rounds. More explanation is needed to justify why the key challenge can be effectively addressed in the first $\sqrt{K}$ rounds.

2. In Lemma 5.2, the Ref-Model is claimed to achieve an approximate transition probability function with a small error $\epsilon_0$. According to the regret analysis in this first stage, $\epsilon_0$ is approximately $1/\sqrt{K}$. However, in general cases, it typically requires $O(1/\epsilon^2)$ rounds to obtain an $\epsilon$-optimal estimator. Thus, it seems unreasonable that the algorithm can achieve a $1/\sqrt{K}$-optimal approximation with only $\sqrt{K}$ rounds. Further clarification is needed on how this estimation is justified.

**Relation To Broader Scientific Literature:**

This work mainly focuses on reinforcement learning with trajectory feedback; however, the proposed algorithm is highly computationally inefficient, making it primarily relevant for the theoretical analysis of reinforcement learning rather than practical applications.

**Theoretical Claims:**

The author provides a clear proof sketch for the case where the transition probability is already known. In this setting, the author utilizes the observation that the number of trajectories is significantly smaller than the number of deterministic policies and represents the reward of each policy as a linear combination of the rewards for each trajectory. By leveraging this method, the author reduces the problem to a linear bandit problem, achieving better performance with a lower-dimensional representation.

However, the correctness of the analysis when the transition probability is unknown is not entirely clear, and I have several concerns:

1. As the author mentions in the regret guarantee, the upper bound matches the lower bound for a more powerful learner with single-step rewards, suggesting that the main challenge in learning an MDP comes from estimating the transition dynamics. Given this, it seems unreasonable that the algorithm in Ref-Model only dedicates a small fraction of rounds to estimating the transition matrix and then primarily focuses on reward estimation using the estimated transition model for the majority of rounds. More explanation is needed to justify why the key challenge can be effectively addressed in the first $\sqrt{K}$ rounds.

2. In Lemma 5.2, the Ref-Model is claimed to achieve an approximate transition probability function with a small error $\epsilon_0$. According to the regret analysis in this first stage, $\epsilon_0$ is approximately $1/\sqrt{K}$. However, in general cases, it typically requires $O(1/\epsilon^2)$ rounds to obtain an $\epsilon$-optimal estimator. Thus, it seems unreasonable that the algorithm can achieve a $1/\sqrt{K}$-optimal approximation with only $\sqrt{K}$ rounds. Further clarification is needed on how this estimation is justified.

---

> ### Author Rebuttal · Authors · 2025-04-01
>
> We are grateful for the reviewer's detailed assessment and helpful suggestions. Below we present our response.
>
>  **Regarding your concerns about correctness:**
> 1. In the first $K_0=O(\sqrt{K})$ episodes (other factors ignored), the main target is to identify the infrequent state-action-state triples with probability $O(\sigma_0)=O(K^{-1/2})$. For the left triples, we want to  compute an  $(3,\sigma_0)$-approximate estimation for the transition probability $P_{s,a,h}$. Note that this approximate transition matrix is only for designing the exploration policy, not for computing the near-optimal policy. Hence, computing such an approximate transition matrix requires only $O(\sqrt{K})$ episodes.
>
> 2. We set $\epsilon_0 = O(K^{-1/4})$ ignoring other factors (please refer to Appendix.A). You are correct that it requires $1/\epsilon_0^2$ episodes to learn an $\epsilon$-optimal policy.
> Indeed, we compute this $\epsilon_0$ optimal policy to improve the efficiency of exploration in the following episodes.
> In the main algorithm, we use $K_0=O(\sqrt{K})$ episodes to identify the triples with probability $O(K^{-1/2})$. If we apply direct exploration, the regret in this stage might be $K_0H$, which violates the minimax bound $O(\sqrt{SAH^3K})$ due to the dependencies on $S,A$ and $H$. Instead, we first compute the set of $\epsilon_0$ optimal policies, and then conduct exploration within this policy set. In this way, the regret due to exploration is bounded by $O(K_1 H + K_0 \epsilon) = O(\sqrt{SAH^3K})$.
>
> **Regarding the computational inefficiency:**
>
> We admit that the current algorithm is computationally inefficient, and indeed, devising algorithm with minimax optimal regret bound and polynomial running time for RL with trajectory feedback is an  intriguing open problem. However, we notice that obtaining statistically (nearly) optimal algorithms is usually the first step for completely settling problems in machine learning theory, and we believe our techniques would be useful for later developments in the theory of RL with trajectory feedback.
>
> **Regarding  the large dependency on $S,A,H$:**
>
> Indeed our current regret bound has large burn-in terms, and we would leave improving those terms as a future direction. On the other hand, we would like stress that even for RL in the standard setting, minimax optimal regret bounds have not been obtained until recent few years, and the burn-in terms were not fully optimized until very recently. Therefore, as the first work that achieves minimax optimal regret bound for RL with trajectory, we believe it is reasonable to have large burn-in terms in our bound. Moreover, we believe our new techniques are crucial for fully understanding the regret bound of RL with trajectory feedback.
>
> **Regarding other comments or suggestions:**
>
> We are sorry for the typos. The $T$ notation in Section 5 denotes number of episodes, not number of steps. We will use $\check{K}$ to replace $T$ throughout Section 5 and the corresponding analysis in the next revision. We will also separate $K$ from other factors. The final regret bound would be $\tilde{O}(\sqrt{SAH^3K}+S^{24}A^4 H^{32})$.

---

> > ### Comment · Reviewer_egr3 · 2025-04-03
> >
> > Thanks for the rebuttal and it addresses my concern regarding the algorithm. As the authors mentioned, the approximate transition matrix is only used for designing the exploration policy. There is a subsequent exploration stage in Algorithm 5, and the update policy is based on newly collected data (Algorithm 5, Line 7).
> >
> > Additionally, in Algorithm 1, all Traj-Learning is based on the reference model computed during the first stage. It is interesting that if we replace it with the estimated transition probability function from Algorithm 5 during the iterative structure, would that help reduce the burn-in terms?
> >
> > Overall, I will maintain my positive score.

---

> > > ### Author Response · Authors · 2025-04-04
> > >
> > > Thanks for the follow-up question.
> > >
> > > **Regarding improving the burn-in terms using new estimated transition matrix:** We think the answer is not. In our current analysis, a $(3,\sigma_{0})$-approximate transition matrix suffices for designing the exploration policy. A better approximation of the transition matrix does not improve the efficiency of policy design.

---

### Decision · Program_Chairs · 2025-05-01

**Decision:**

Accept (poster)

**Comment:**

We thank the authors for their submission.
All in all, the contributions were clear, the paper is well-written and the proofs appear to be sound.
The reviews were reserved about the algorithm not being computationally efficient, and excessively large regret incurred during the burn-in period. Perhaps these can be studied in future work.